# Effects of Intercropping *Pandanus amaryllifolius* on Soil Properties and Microbial Community Composition in *Areca Catechu* Plantations

Yiming Zhong [1,2,3], Ang Zhang [1,2,3,*], Xiaowei Qin [1,2,3], Huan Yu [1,2,3], Xunzhi Ji [1,2,3], Shuzhen He [1,2,3], Ying Zong [1,2,3], Jue Wang [1,2,3] and Jinxuan Tang [1,2,3]

1 Spice and Beverage Research Institute, Chinese Academy of Tropical Agricultural Sciences, Wanning 571533, China
2 Hainan Provincial Key Laboratory of Genetic Improvement and Quality Regulation for Tropical Spice and Beverage Crops, Spice and Beverage Research Institute, Chinese Academy of Tropical Agricultural Sciences, Wanning 571533, China
3 Key Laboratory of Genetic Resource Utilization of Spice and Beverage Crops of Ministry of Agriculture and Rural Affairs, Spice and Beverage Research Institute, Chinese Academy of Tropical Agricultural Sciences, Wanning 571533, China
* Correspondence: angzhang_henu@163.com

**Abstract:** The areca nut (*Areca catechu* L.) and pandan (*Pandanus amaryllifolius* Roxb.) intercropping cultivation system has been widely practiced to improve economic benefits and achieve the development of sustainable agriculture in Hainan Province, China. However, there is a lack of research on the relationships among soil properties, soil enzyme activities, and microbes in this cultivation system. Therefore, a random block field experiment of pandan intercropped with areca nut was established to investigate the effects of environmental factors on the diversity and functions of soil microbial communities in Lingshui county, Hainan Province. The diversity and composition of soil microbial communities under different cropping modes were compared using Illumina sequencing of 16S rRNA (bacteria) and ITS-1 rRNA (fungi) genes, and FAPROTAX and FUNGuild were used to analyze and predict the bacteria and fungi community functions, respectively. Correlation analysis and redundancy analysis were used to explore the responses of soil microbial communities to soil environmental factors. The results showed that the bacterial community was more sensitive to the areca nut and pandan intercropping system than the fungal community. The functional predictions of fungal microbial communities by FAPROTAX and FUNGuild indicated that chemoheterotrophy, aerobic chemoheterotrophy, and soil saprotroph were the most dominant functional communities. The intercropping of pandan in the areca nut plantation significantly enhanced the soil bacterial Ace and Chao indices by reducing the soil organic carbon (SOC) and total phosphorus (TP) content. In the intercropping system, urease (UE) and acid phosphatase were the key factors regulating the soil microbial community abundance. The dominant bacterial and fungal phyla, such as Firmicutes, Methylomirabilota, Proteobacteria, Actinobacteria, Chloroflexi, Verrucomicrobia, and Ascomycota significantly responded to the change in planting modes. Soil properties, such as UE, total nitrogen, and SOC had a significant stimulating effect on Proteobacteria, Chloroflexi, and Ascomycota. In summary, soil bacteria responded more significantly to the change in cropping modes than soil fungi and better reflected the changes in soil environmental factors, suggesting that intercropping with pandan positively affects soil microbial homeostasis in the long-term areca nut plantation.

**Keywords:** cultivation mode; soil physicochemical properties soil enzyme activity; soil microbial diversity; microbial community structure



## 1. Introduction

With the development of large-scale and intensive agricultural production modes, the degradation of farmland soil microbial communities has become a prominent problem

that restricts the sustainable development of agriculture. As a farming mode based on the principle of promoting and complementing ecology, intercropping allows two or more crops to be planted on the same field, thereby, not only optimizing the utilization of resources such as sunlight, water, nutrients, and shared space [1], improving the net effect in the tradeoff between interspecies competition and the facilitation of crop growth [2,3], and considerably increasing yield [4], but also stimulating the interactions among soil nutrients, enzymes, microbes, and several coexisting crops [5], thus, maintaining the relative balance of soil microbial community [6,7]. Areca nut is an important cash crop in the tropical regions of South and Southeast Asia [8,9], which is often intercropped with vegetables, cocoa, banana, black pepper, and cardamom [10]. Among them, pandan is a tropical spice crop with high economic value; it is shade-tolerant and suitable for intercropping in areca nut forests [11–13]. Therefore, exploring the relationship among soil properties, enzyme activities, and microbes, as well as the mechanism of the intercropping mode to maintain soil health, is conducive to maintaining the efficient production of areca nut and pandan. Soil enzyme activity is a vital indicator of soil quality and is essential in evaluating soil health [14]. Common soil enzymes, such as catalase (CAT), acid phosphatase (ACP), urease (UE), and invertase, play a catalytic role in the decomposition of plant and animal residues, accelerating their biochemical reactions [15]. UE participates in the ammoniation of organic nitrogen in the nitrogen cycle in the farmland ecosystem to produce plant-available nitrogen [16]. UE activity determines the transfer rate of soil nutrients [17]. Peroxidase (POD) degrades lignin and coupled polysaccharides and is related to the degradation of polyphenols produced by soil fungi [18,19].

Microbial community composition is related to soil function and ecosystem sustainability because it is involved in soil organic matter dynamics and nutrient cycling processes, as well as in the metabolism of the soil system [20–22]. Soil microbes are diverse and functionally valuable, containing various species of bacteria, archaea, fungi, microfauna, and viruses [23]. As the two major categories of the farmland soil microbial system, bacteria and fungi usually represent the soil microbial community and are used to analyze and compare the soil microbial diversity indices and community structure [24]. Soil management measures can directly affect soil properties by changing the relationship between soil microbial community and soil properties [21]. For example, planting modes affect the development and vitality of soil microbes, mainly by changing soil properties. Lower pH affects bacterial and fungal densities. A moderate improvement of soil organic carbon (SOC) and soil nitrogen content stimulate soil microbial abundance and diversity [25], whereas excessive nutrient addition inhibits microbial diversity. There is a close relationship between soil enzyme activities and soil microbial characteristics because soil microbes are capable of secreting a range of enzymes, and changes in soil enzyme activities reflect changes in the nutrient requirements and metabolic activity of soil microbes [26]. Intercropping affects the relationship between soil enzyme activity and the microbial community by changing soil properties and microenvironments and then regulates the structure and function of the soil microbial community [27]. It is noteworthy that the intercropping of different crops has various effects on soil microbial content: intercropping of Kura clover with prairie cordgrass increases the abundance of arbuscular mycorrhizal fungi [28], intercropping wolfberry with Gramineae plants increases bacterial alpha diversity [29], and a melon/cowpea intercropping system enhanced the content of beneficial microbes [2]. Legumes and nitrogen fixation may increase the nitrogen content of the soil, but in other intercropping systems, different effects may occur [30].

Soil microbial diversity is inextricably linked to microbial function, and increased microbial diversity implies the improved soil biochemical response and sustainability of soil function [31]. Previous studies considered that the major functions of soil microbes are regulating soil functional diversity [32], decomposing plant residues [22], maintaining soil fertility and productivity, participating in carbon and nitrogen cycling [33], and inhibiting pathogens [34]. The effect of intercropping on soil microbial diversity also can significantly alter soil function [35]. Complex interactions exist among soil resources, soil enzymes,

and soil microbes in the intercropping system [5]. Intercropping causes changes in soil properties and nutrients, significantly affecting the metabolic activities of soil microbes [6], including the production of cellulase that decomposes polysaccharides, UE, ACP [36], and neutral and alkaline phosphatases that participate in nitrogen and phosphorus cycles [37]. Moreover, the interactions among microbes, soil nutrients, and various enzymes in the intercropping system cause changes in the abundance of microbes and enzyme activities, which can improve the soil micro-ecological environment and functions [38].

However, the effects of intercropping of cash crops on soil microbial communities are still unclear in tropical farmland. Therefore, the areca nut and pandan intercropping field experiment was established to: (1) clarify the effects of intercropping on soil properties, enzyme activities, and microbial community diversity and structure; (2) explore the key mechanism of how the intercropping system alters soil microbial community diversity and structure; and (3) investigate the effect of soil microbial community functional change in a tropical intercropping system.

## 2. Materials and Methods

### 2.1. Study Site

The experiment was performed in Sanjiaowei Village, Lingshui County (109°56′ E, 18°31′ N, a.s.l. 36 m) in southeastern Hainan Province, China, from 2015. The mean annual temperature was 25 °C and the mean annual precipitation was 1700 mm at the experimental site. The soil was tidal sand–mud (US Soil Taxonomy classification) with a pH of 6.00, organic matter of 20.04 g·kg$^{-1}$, electrical conductivity (EC) of 96.68 S·m$^{-1}$, and soil-available nitrogen (SAN), soil-available phosphorus (SAP), and soil-available potassium (SAK) concentrations of 77.78 mg·kg$^{-1}$, 17.22 mg·kg$^{-1}$, and 51.46 mg·kg$^{-1}$, respectively. Total nitrogen (TN), total phosphorus (TP), and total potassium (TK) were 1.33 g·kg$^{-1}$, 0.82 g·kg$^{-1}$, and 11.93 g·kg$^{-1}$, respectively.

### 2.2. Experimental Design and Management

The experiment adopted a randomized block design. Each block had three plots and each block was replicated 6 times, and one plot was set for each planting mode: areca nut monocropping (AM), pandan monocropping (PM), and areca nut and pandan intercropping (I), and the block was repeated 6 times. The cultivation period of areca nut is about 6 years, and the cultivation period of fragrant pandan is about 3 years. The planting density was 2.5 m × 2.5 m for areca nut and 50 cm × 50 cm for pandan. During the experiment, water and fertilizer management, pest control, and other field management practices remained the same in the three planting modes.

### 2.3. Soil Sampling

Soil samples were collected in June 2021. Five soil samples (0–20 cm) were randomly collected from each plot with a diameter of 5 cm and then mixed as one soil sample. After sieving (<2 mm, <0.20 mm) to remove plant roots and other visible foreign bodies, all soil samples were immediately brought back to the laboratory. Soil samples were divided into two parts: one was used to analyze soil physicochemical properties after air drying, and the other was stored in a −80 °C freezer for soil microbial community analysis.

### 2.4. Analysis of Soil Physicochemical Properties and Soil Enzyme Activities

Soil pH was measured using a pH/conductivity meter (FE28, China; soil: water ratio was 1:2.5). After weighing the fresh weight, the soil samples were oven-dried at 105 °C for 24 h and weighed again to calculate the soil water content (SWC) [39]. Electrical conductivity (EC) was measured using the pH/conductivity meter (DDS-307A conductivity meter, China) [40]. Soil organic matter was determined by a total organic carbon analyzer (Multi N/C 3100, Jena, Germany) [41], and bulk density (BD) was measured (BD, g/cm$^3$ = soil dry weight/soil volume). Alkali-hydrolyzed nitrogen (SAN) was determined using the alkaline hydrolysis diffusion method. SAP was assessed using Bray's method (UV2310 II, Shanghai,

China) [42]. SAK was determined using flame photometry (6400A, Changsha, China) [43]. TN was determined by Kelvin distillation, TP using the molybdenum blue colorimetric method, and TK by flame photometry [44]. Soil catalase (CAT), soil polyphenol oxidase (PPO), soil peroxidase (POD), soil acid phosphatase (ACP), and soil urease (UE) were determined by ultraviolet–visible spectrophotometry using kits (Suzhou Comin Biotechnology Co., Ltd., Suzhou, China). PPO can catalyze pyrogallol to produce colored species with characteristic light absorption at 430 nm. $H_2O_2$ has a characteristic absorption peak at 240 nm. By measuring the change in the absorbance of the solution at this wavelength after reacting with the soil, the activity level of CAT can be reflected. POD catalyzes the oxidation of organic substances to quinones, which have characteristic light absorption at 430 nm. Using the indophenol blue colorimetric method, the $NH_3$-N generated by the urease hydrolysis of urea was identified (www.cominbio.com).

### 2.5. Soil DNA Extraction and Sequencing

Total soil DNA was extracted and purified using the EZNA® Soil DNA Extraction Kit (Omega, Norwalk, CT, USA). Using barcode-tagged primer sequences for bacteria: 338F (5′-ACTCCTACGGGAGGCAGCAG-3′) and 806R (5′-GGACTACHVGGGTWTCTAAT-3′), and fungi: internal transcribed spacer (ITS) 1F (5′-CTTGGTCATTTAGAGGAAGTAA-3′) and ITS2R (5′-GCTGCGTTCTTCATCGATGC-3′), the corresponding soil bacterial 16S rRNA V3-V4 region and fungal ITS-1 region sequences were amplified, and 2% agarose gel electrophoresis was used to detect the length of the amplified products. According to the quantitative detection results, the amplified products were mixed into one sample, and a clone library was constructed. The loading amount for each library was calculated based on the library search results, and the paired-end sequencing method was used on the Illumina MiSeq high-throughput platform for sequencing. The data were analyzed using the Majorbio cloud platform (www.Majorbio.com (accessed on 22 September 2022)).

### 2.6. Bioinformatics Analysis

Paired-end reads of raw DNA fragments were merged using FLASH 1.2.11 [45] software and quality filtered using QIIME 1.9.1 software [46]. Valid sequences were obtained, and reads that could not be assembled were discarded. Unique sequences with 97% or greater similarity were clustered into operational taxonomic units (OTUs) using UPARSE 7.0.1090 software. MOTHUR 1.30.2 [47] annotated each OTU using the small subunit rRNA SILVA database (v 138) [48] and UNITE 8.0 fungi database [49]. The sample with the least data was used as the standard for normalization (normalization using the normalization method: the sequences of all samples are randomly selected to that amount of data according to the minimum number of sample sequences). Soil microbial community diversity and richness were calculated using QIIME.

### 2.7. Statistical Analysis

Taxonomic alpha diversity was calculated as the estimated community diversity by the Shannon index using the Mothur software package (v.1.30.1), and nonmetric multidimensional scaling (NMDS) was selected to reflect the changes in the microbial structure under intercropping modes, these changes were referred to as microbial beta diversity. Network interaction analysis of microbial composition was analyzed and painted by SPSS 23.0 and Cytoscape V3.8.2, respectively. FAPROTAX (v1.2.1) [50] and FUNGuild (v1.0) [51] were used to analyze and predict the microbial community functions, respectively. The same community with different guild annotations was selected for all annotations in FUNGuild using three classification levels: highly probable, probable, and possible. The microbial community was divided into bacteria and fungi in this study (after confirmation, no archaeal taxa were found in the 16 s dataset). The experimental indicator (soil physical and chemical properties, soil enzyme activity, soil fungal–bacterial diversity and community structure, and prediction of soil fungal–bacterial functional communities) was analyzed by one-way ANOVA to determine differences between intercropping and monocropping

modes. NMDSs were statistically assessed using a permutational analysis of variance (PERMANOVA). The statistical significance ($p < 0.05$) was calculated using Duncan's test. Correlations between soil properties, soil enzyme activities, and soil microbial community diversity were calculated and analyzed using a Spearman correlation matrix. Redundancy analysis was performed and mapped using the analysis of soil microbial community composition about environmental factors; the model was assessed for 999 iterations based on Monte Carlo permutations. Data analyses were performed using SAS V8 and Canaco 5.0. The graphs were plotted using Origin 2021b and R.4.0.5.

## 3. Results

### 3.1. Changes in Soil Physicochemical Properties and Enzyme Activities

One-way ANOVA revealed the significant effects of planting modes on soil properties and enzyme activities. Compared to the AM mode, intercropping significantly increased pH, BD, and TK by 0.47 (absolute difference, $p < 0.001$), 16.71% (absolute difference, $p < 0.001$), and 18.44% (relative difference, $p < 0.05$), respectively, whereas EC, SOC, SAK, SAN, SOP, TN, and TP were significantly decreased by 41.52% ($p < 0.001$), 22.28% ($p < 0.001$), 47.80% ($p < 0.001$), 36.98% ($p < 0.001$), 23.79% ($p < 0.01$), 22.33% ($p < 0.001$), and 47.24% ($p < 0.01$) under the intercropping mode, respectively (Figure 1). Most of the soil physical and chemical properties were significantly lower under the intercropping mode when compared with PM monoculture ($p < 0.01$), except the soil TK content was significantly increased by 9.94% ($p < 0.001$), and SWC, SAK, and SAN were not affected. Compared to the AM mode, intercropping significantly increased ACP and UE activity by 36.64% and 8.27% (relative difference, $p < 0.01$). However, the activity of CAT, PPO, and UE were significantly decreased by 14.61%, 37.07%, and 20.04% when compared with PM, while POD and ACP in the intercropping mode were significantly higher than in the PM mode by 70.64% and 33.48%, respectively (Figure 2). There is a gigantic difference between the AM and PM modes in the physicochemical characteristics and enzyme activity. The soil properties and enzyme activities of the areca nut forest were altered dramatically after intercropping with pandan in this study.

### 3.2. Changes in Soil Microbial Community Diversity

The number of soil bacterial community sequences per sample ranged from 29,519 to 62,515 (mean = 41,633), whereas the number of fungal community sequences ranged from 54,987 to 87,289 (mean = 66,394). The intercropping mode significantly increased the bacterial Ace and Chao indices by 28.30% and 27.26% (relative difference), whereas other soil microbial diversity indices did not change significantly (Figure 3). One-way ANOVA revealed that intercropping did not affect the Shannon or Simpson index when compared with AM or PM, whereas it significantly increased Ace and Chao indices by 28.24% and 28.67%, respectively, in the bacterial community when compared with PM. When compared with AM, intercropping increased the bacterial Shannon index by 5.08%. Ace and Chao indices were increased by 72.17% and 69.95%, respectively, in the fungal community after intercropping, when compared with PM ($p < 0.05$, Figure 3).

Nonmetric multidimensional scaling analysis (NMDS) was conducted to reflect microbial beta diversity (Appendix A, Figure A1). The soil bacterial characteristics for the AM and intercropping treatments were nearly the same, whereas the soil bacterial characteristics under PM treatment were quite different from the intercropping AM mode. The soil fungal characteristics were not greatly affected by the intercropping modes in this study (Appendix A, Figure A1).

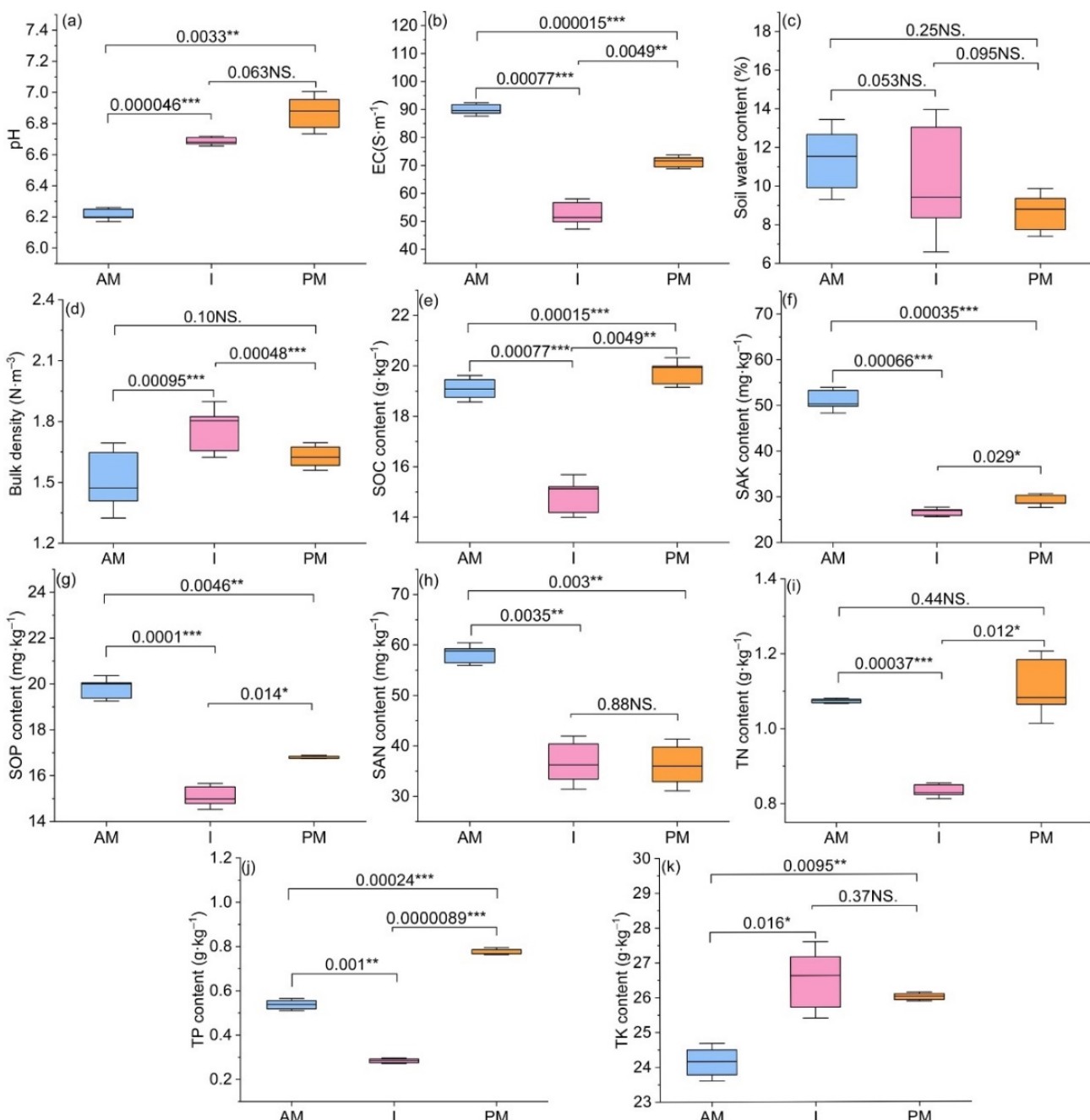

**Figure 1.** Soil properties under different cropping patterns (n = 9). * is significant at the 0.05 level; ** is significant at the 0.01 level; and *** is significant at the 0.001 level. AM represents areca nut monocropping; I represents areca nut intercropping with pandan; and PM represents pandan monocropping. Note: (**a**)-pH, (**b**)-EC, (**c**)-SWC, (**d**)-BD, (**e**)-SOC, (**f**)-SAK, (**g**)-SOP, (**h**)-SAN, (**i**)-TN, (**j**)-TP, (**k**)-TK.

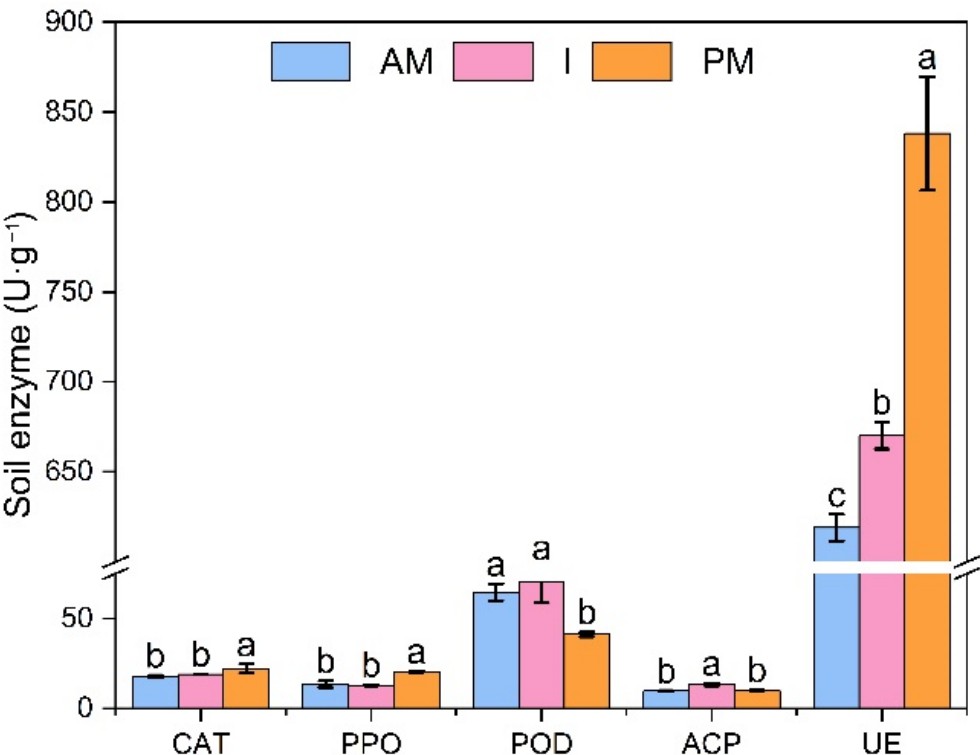

**Figure 2.** Soil enzyme activity under different cropping patterns. Different letters indicate significant differences (ANOVA, $p < 0.05$, and Tukey's HSD post hoc analysis) among different planting modes. AM represents areca nut monocropping; I represents areca nut intercropping with pandan; and PM represents pandan monocropping. Different lowercase letters indicate significant differences between treatments under the same index ($p < 0.05$).

### 3.3. Changes in the Composition and Structure of Soil Microbial Community

The phyla with relative abundance greater than 1% in soil bacterial and fungal communities are usually considered the dominant phyla. The 12 dominant phyla in the bacterial community were Proteobacteria (25.44%), Actinobacteria (20.94%), Acidobacteria (15.64%), Firmicutes (10.40%), Chloroflexi (7.89%), Bacteroides (4.24%), Myxococcota (3.24%), Methylomirabilota (1.96%), Verrucomicrobia (1.59%), Gemmatimonadota (1.25%), Planctomycetota (1.13%), and Bdellovibrionota (1.07%). The four dominant phyla in the fungal community were Ascomycota (76.77%), Basidiomycota (11.33%), unclassified fungi (7.62%), and Rozellomycota (2.75%) (Figure 4). Compared with AM, Firmicutes in intercropping significantly decreased by 12.61%, whereas Methylomirabilota and unclassified bacteria were significantly increased by 2.88% and 0.68%, respectively, and Acidobacteria abundance increased by 5.86%. Compared with PM, Proteobacteria, Ascomycota, and Chloroflexi were significantly reduced by 1.62%, 16.45%, and 1.89%. Methylomirabilota and Verrucomicrobia were significantly increased by 1.35% and 1.93% (absolute difference, all $p < 0.05$), respectively, after intercropping (Figure 4, Appendix A, Table A1). There was a strong positive correlation among the dominant bacteria groups: Acidobacteriota, Actinobacteriota, Bacteroidota, Chloroflexi, Firmicutes, Methylomirabilota, Myxococcota, and Proteobacteria. However, the dominant fungal community Ascomycota showed a strong negative correlation with other fungal groups except Zoopsgomycota (Figure 5).

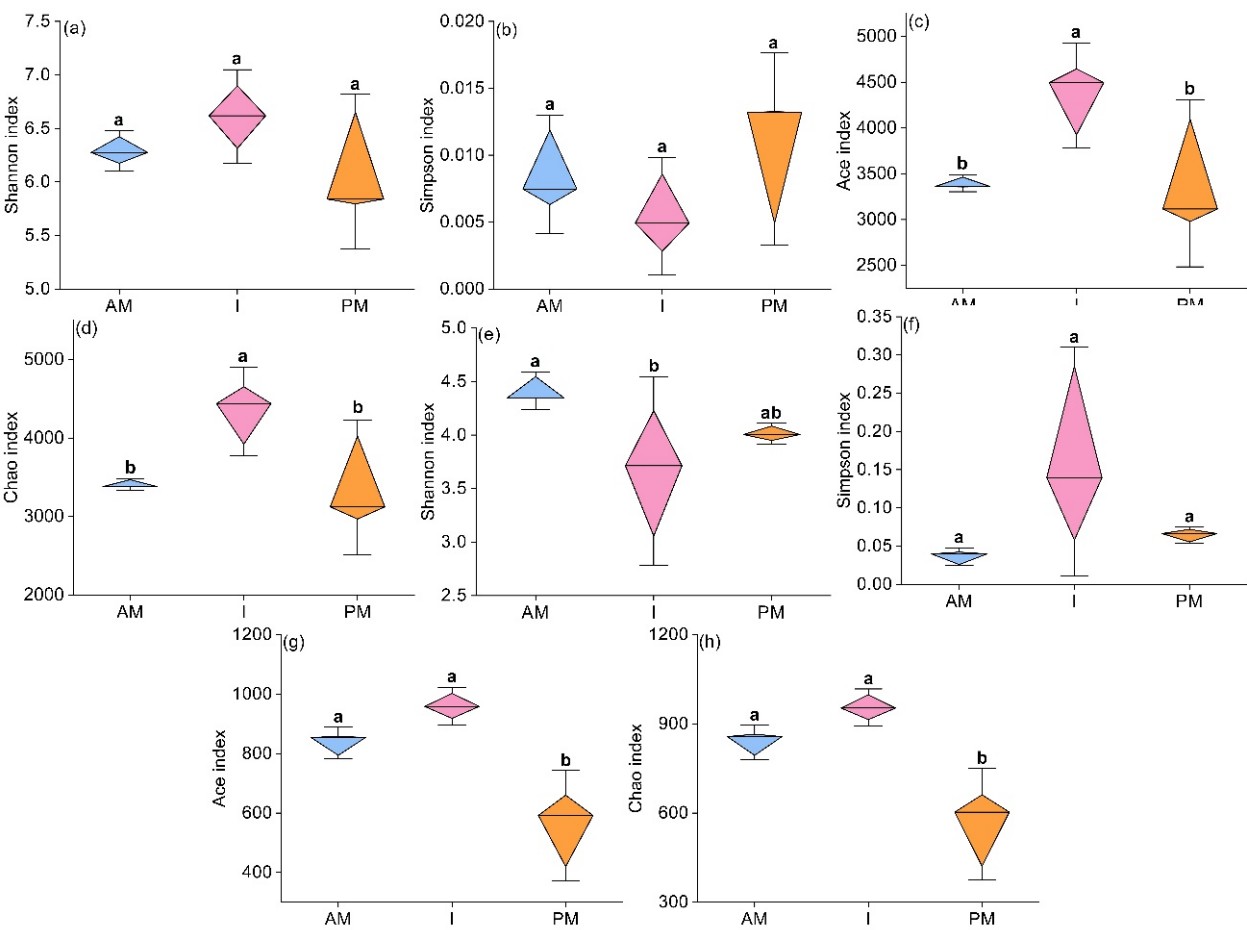

**Figure 3.** Changes in soil microbial alpha diversity index under different planting patterns ((**a**–**d**): bacteria; (**e**–**h**): fungi). Different letters indicate significant differences (ANOVA, $p < 0.05$, and Tukey's HSD post hoc analysis) among different planting modes. AM represents areca nut monocropping; I represents areca nut intercropping with pandan; and PM represents pandan monocropping. Different lowercase letters indicate significant differences between treatments under the same index ($p < 0.05$).

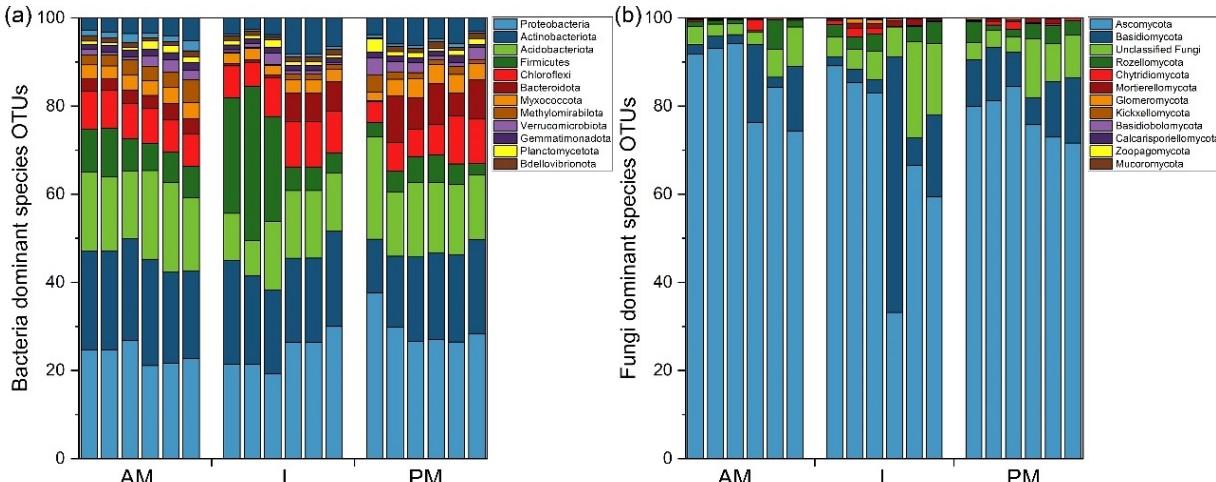

**Figure 4.** Soil microbial community composition under different planting patterns ((**a**)-bacteria, (**b**)-fungi). The abundance of each taxon was calculated as the percentage of sequences per gradient for a given microbial group. AM represents areca nut monocropping; I represents areca nut intercropping with pandan; and PM represents pandan monocropping.

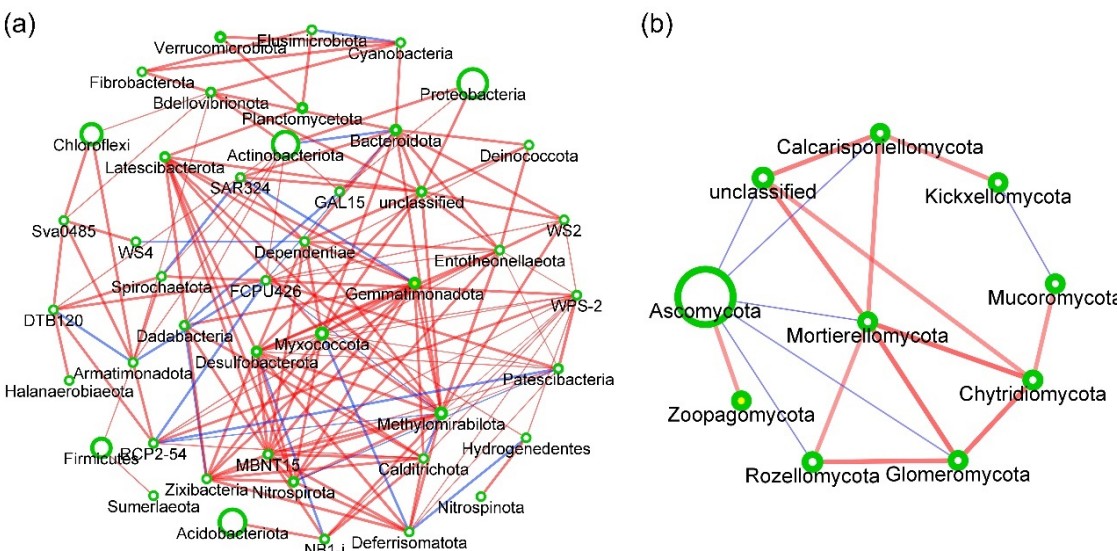

**Figure 5.** Network interaction diagram of dominant bacterial (**a**) and fungal (**b**) phyla. The line between the circles indicates that there is a correlation, red lines indicate a positive correlation, while the blue lines show a negative correlation. The size of the points represents the magnitude of phyla abundance, while the thickness of the line represents the correlation size. Each circle represents a microbial phylum. Red indicates a positive correlation, and blue indicates a negative correlation ($p < 0.05$).

### 3.4. Changes in Soil Microbial Functional Profiles

The functional prediction of the soil bacterial community showed that the main functional groups in each plot were "chemoheterotrophy" and "aerobic chemoheterotrophy". The majority of bacteria in nature are chemoheterotrophic bacteria, and their energy comes from the oxidation and decomposition of soil organic matter. The relative content of chemoisomeric bacteria in the AM and I modes was significantly lower than in the PM (Figure 6a, Appendix A, Table A2). The main functional prediction of the soil fungal community was "soil saprotroph". Soil saprophytic fungi absorb nutrients from dead plant residues or other organic substances, which are also chemoautotrophic microbes in nature. The relative abundance of soil saprotroph fungi in the PM treatment was slightly lower than that in the AM and I treatments in this study. It was worth noting that the relative abundance of "Symbiotroph" in fungi under the intercropping treatment was significantly lower than that in the AM or PM treatment (Figure 6b, Appendix A, Table A3).

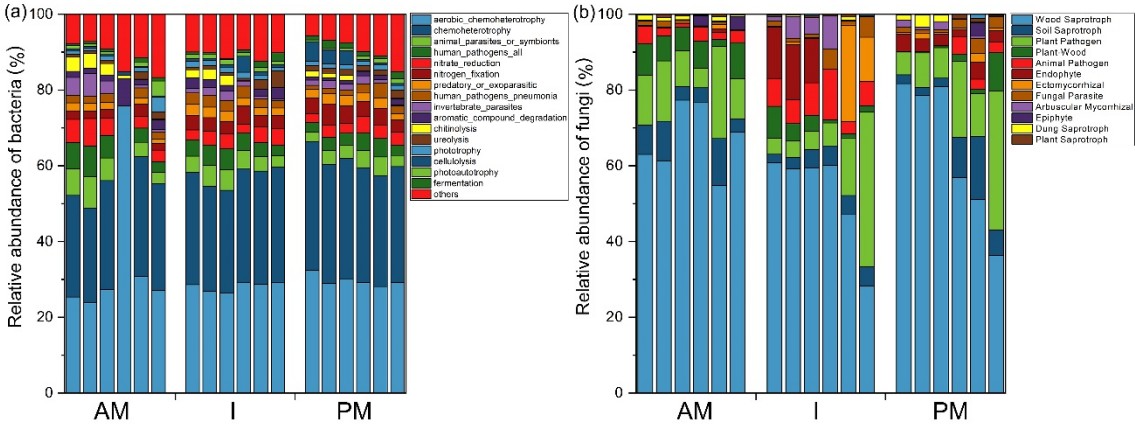

**Figure 6.** Predicted functional profiles of the soil bacteria (**a**) and fungi (**b**) with different planting mode. AM represents areca nut monocropping; I represents areca nut intercropping with pandan; and PM represents pandan monocropping.

### 3.5. Relationship between Soil Properties and Soil Enzymes

A close relationship was observed between soil properties and enzyme activities. There was a positive and negative correlation between TP ($R = 0.87$), SWC ($R = -0.66$), and PPO, respectively. POD was significantly negatively correlated with TP ($R = -0.84$). ACP was highly significantly negatively correlated with SOC ($R = -0.96$), TN ($R = -0.91$), EC ($R = -0.84$), and TP ($R = -0.82$), SOP ($R = -0.78$), whereas it was positively correlated with BD. UE was significantly positively correlated with TP ($R = 0.71$) and pH ($R = 0.85$). However, CAT was not correlated with soil physicochemical properties (all $p < 0.05$, Figure 7).

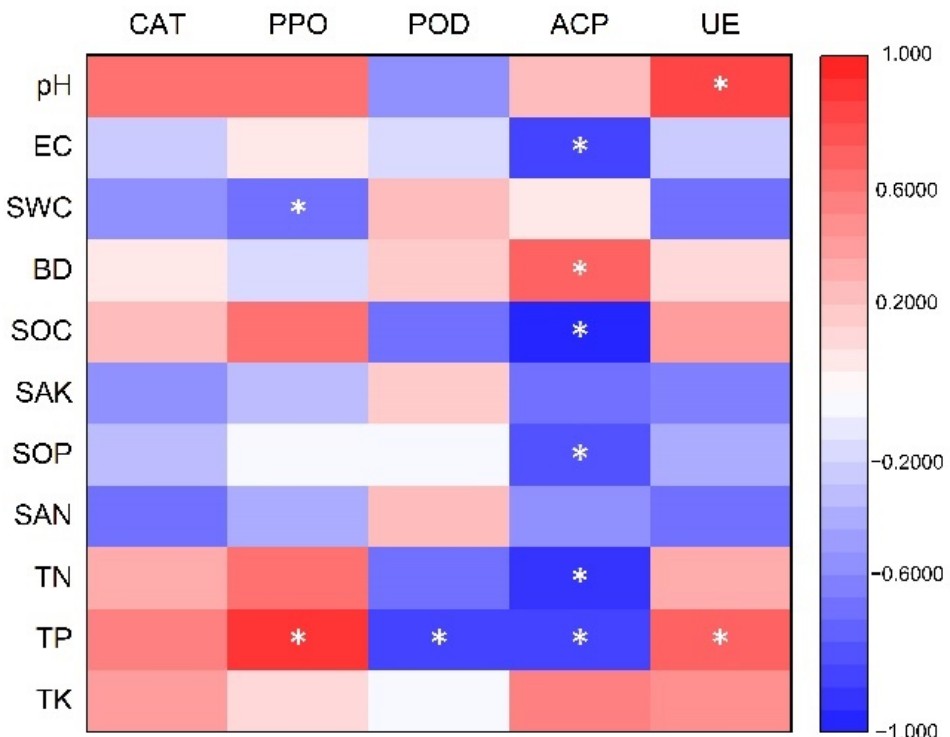

**Figure 7.** Relationship between soil environmental factors and soil enzyme activities. * Correlation is significant at the 0.05 level. Red indicates a positive correlation and blue indicates a negative correlation, and the darker the color, the stronger the correlation. Electrical conductivity (EC), soil water content (SWC), soil bulk density (BD), soil organic carbon (SOC), soil-available potassium (SAK), soil-available phosphorus (SAP), alkali-hydrolyzed nitrogen (SAN), total nitrogen (TN), total phosphorus (TP), total potassium (TK), soil catalase (CAT), soil polyphenol oxidase (PPO), soil peroxidase (POD), soil acid phosphatase (ACP), and soil urease (UE). Same below.

### 3.6. Influence of Soil Biological and Abiotic Factors on Soil Microbial Community Diversity

The Ace and Chao indices of the bacterial community were negatively correlated with SOC ($R = -0.79$, $-0.80$; $p < 0.05$, 0.01) and TP ($R = -0.71$, $-0.73$; $p < 0.05$), respectively, but positively correlated with ACP ($R = 0.83$, 0.84, $p < 0.01$). Fungal Ace and Chao indices were negatively correlated with SOC ($R = -0.75$, $-0.74$; $p < 0.05$), TN ($R = -0.68$, $-0.67$; $p < 0.05$), TP ($R = -0.91$, $-0.91$; $p < 0.001$), CAT ($R = -0.69$, $-0.70$; $p < 0.05$), PPO ($R = -0.91$, $-0.92$; $p < 0.001$), and UE ($R = -0.84$, $-0.84$; $p < 0.01$), respectively, but positively correlated with POD ($R = 0.88$, 0.88; $p < 0.01$). The fungal Shannon index was positively correlated with EC ($R = 0.67$; $p < 0.05$), SAK ($R = 0.68$; $p < 0.05$), and SOP ($R = 0.73$; $p < 0.05$), respectively, but negatively correlated with ACP ($R = -0.76$; $p < 0.05$). The fungal Simpson index was positively correlated with ACP ($R = 0.82$; $p < 0.01$, Table 1).

**Table 1.** Relationship between soil microbial alpha diversity and environmental factors.

| Soil Properties | Bacteria | | | | Fungi | | | |
|---|---|---|---|---|---|---|---|---|
| | Shannon | Simpson | Ace | Chao | Shannon | Simpson | Ace | Chao |
| pH | −0.19 | 0.17 | 0.12 | 0.10 | −0.53 | 0.30 | −0.45 | −0.46 |
| EC | −0.31 | 0.28 | −0.64 | −0.63 | 0.67 * | −0.60 | −0.26 | −0.24 |
| SWC | 0.05 | −0.03 | −0.02 | 0.00 | 0.16 | −0.02 | 0.48 | 0.49 |
| BD | 0.28 | −0.16 | 0.60 | 0.59 | −0.58 | 0.59 | 0.26 | 0.25 |
| SOC | −0.61 | 0.58 | −0.79 * | −0.80 ** | 0.56 | −0.65 | −0.75 * | −0.74 * |
| SAK | −0.16 | 0.14 | −0.48 | −0.46 | 0.68 * | −0.53 | 0.11 | 0.12 |
| SOP | −0.31 | 0.27 | −0.64 | −0.63 | 0.73 * | −0.65 | −0.15 | −0.14 |
| SAN | −0.09 | 0.06 | −0.39 | −0.37 | 0.54 | −0.37 | 0.21 | 0.22 |
| TN | −0.42 | 0.38 | −0.64 | −0.66 | 0.59 | −0.66 | −0.68 * | −0.67 * |
| TP | −0.63 | 0.57 | −0.71 * | −0.73 * | 0.35 | −0.53 | −0.91 *** | −0.91 *** |
| TK | 0.27 | −0.25 | 0.62 | 0.61 | −0.40 | 0.35 | 0.36 | 0.34 |
| CAT | −0.11 | 0.09 | −0.05 | −0.09 | −0.16 | −0.02 | −0.69 * | −0.70 * |
| PPO | −0.50 | 0.45 | −0.47 | −0.50 | 0.04 | −0.27 | −0.91 *** | −0.91 *** |
| POD | 0.50 | −0.50 | 0.52 | 0.54 | 0.07 | 0.13 | 0.88 ** | 0.88 ** |
| ACP | 0.64 | −0.59 | 0.83 ** | 0.84 ** | −0.76 * | 0.82 ** | 0.64 | 0.63 |
| UE | −0.48 | 0.45 | −0.30 | −0.33 | −0.22 | −0.03 | −0.84 ** | −0.84 ** |

Note: * Correlation is significant at the 0.05 level; ** Correlation is significant at the 0.01 level; and *** Correlation is significant at the 0.001 level. Electrical conductivity (EC), soil water content (SWC), soil bulk density (BD), soil organic carbon (SOC), soil-available potassium (SAK), soil-available phosphorus (SAP), alkali-hydrolyzed nitrogen (SAN), total nitrogen (TN), total phosphorus (TP), total potassium (TK), soil catalase (CAT), soil polyphenol oxidase (PPO), soil peroxidase (POD), soil acid phosphatase (ACP), and soil urease (UE).

### 3.7. Responses of Soil Microbial Community Structure to Three Planting Modes

The soil TP ($F$ = 6.6, $p$ = 0.004) and pH ($F$ = 5.5, $p$ = 0.012) significantly affected the soil bacterial community structure in this study (Figure 8a). Soil enzyme activities such as POD ($F$ = 3.6, $p$ = 0.022) had significant effects on soil bacteria (Appendix A, Table A5).

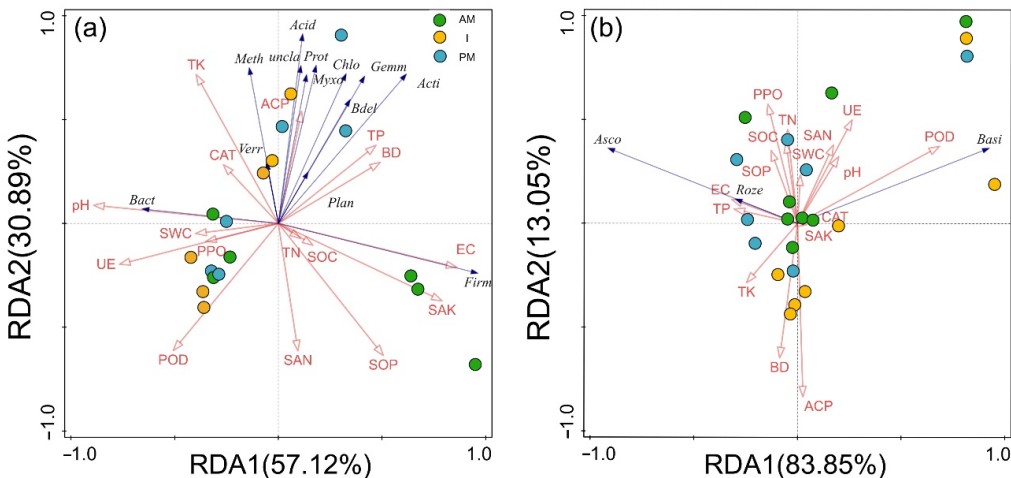

**Figure 8.** Ordination plots of the results from the redundancy analysis (RDA) to identify the relationships among the microbial (bacterial and fungal) taxa (blue arrows) and the soil properties and enzyme activities (red arrows) at the phylum level. (**a**) Relationships between soil bacterial communities and environmental variables and (**b**) relationships between soil fungal communities and environmental variables. Bacterial taxa: Proteobacteria (Prot), Actinobacteriota (Acti), Acidobacteriota (Acid), Firmicutes (Firm), Chloroflexi (Chlo), Bacteroidota (Bact), Myxococcota (Myxo), Methylomirabilota (Meth), Verrucomicrobiota (Verr), Gemmatimonadota (Gemm), Planctomycetota (Plan), and Bdellovibrionota (Bdel). Fungal taxa: Ascomycota (Asco), Rozellomycota (Roze), and Basidiomycota (Basi). Soil properties: electrical conductivity (EC), soil water content (SWC), soil bulk density (BD), soil organic carbon (SOC), soil-available potassium (SAK), soil-available phosphorus (SAP), alkali-hydrolyzed nitrogen (SAN), total nitrogen (TN), total phosphorus (TP), total potassium (TK), soil catalase (CAT), soil polyphenol oxidase (PPO), soil peroxidase (POD), soil acid phosphatase (ACP), and soil urease (UE). Same below.

Most bacterial phyla, such as Proteobacteria and Actinobacteria, were negatively correlated with TN, SOC, PPO, and POD, respectively. Ascomycota was negatively correlated with SOC, SAK, and TN, respectively (Figure 8b). The results of the multiple stepwise regression analysis of the soil microbial phyla indicated that TK, TP, TN, SOC, SAK, EC, UE, and pH were the main factors affecting the abundance of soil bacterial communities such as Actinobacteria, Acidobacteria, and Ascomycota. Among these, the most significant factors affecting soil microbes were TK and SOC (Appendix A, Table A4). Soil fungi community structure was not affected by soil physicochemical properties and soil enzyme activities (Appendix A, Table A5).

## 4. Discussion

### 4.1. Effects of Planting Modes on Soil Enzyme Activities

Soil enzymes are biologically active substances and catalysts involved in soil biochemical processes such as organic matter decomposition and nutrient cycling [52]. The enhancement of soil enzyme activities can accelerate the transformation of organic nutrients and improve the utilization efficiency of nutrients. However, the activities of different soil enzymes differ based on the different crops and planting modes, and the response of soil enzyme activities to soil management varies greatly [29]. In general, researchers consider that soil enzymes were increased with intercropping in the chestnut – tea or cereal–legume intercropping systems [15,53]. However, a meta-analysis indicated that intercropping had an increase, decrease, or neutral effect on soil enzyme activities in most intercropping systems [54].

Specifically, soil ACP catalyzes the mineralization of SOP compounds into inorganic phosphorus, and its activity directly affects SOP decomposition, transformation, and bioavailability [55]. A significant negative correlation was observed between ACP and SOP and TP, indicating that the demand for phosphorus significantly increases in crops under the intercropping mode, thus, stimulating the activity of ACP in the present study (Figure 7). The increase in ACP activity is conducive to the turnover of phosphorus between plants and soil. The soil UE is usually related to the soil nitrogen cycle, and it hydrolyzes urea into ammonia for plant utilization [56,57]. Compared with the AM mode, the soil UE activity of PM was significantly increased, but the soil alkaline-hydrolyzed nitrogen content was significantly reduced, indicating that the demand for nitrogen might be significantly higher in pandan than in areca nut. The soil CAT is mainly related to the degradation of hydrocarbons and heavy metals in soil, and PPO decomposes organic matter and accelerates soil humification [58]. Compared with the PM mode, the decreased CAT and PPO activities under intercropping indicated that the intensity of SOC metabolism (mineralization) significantly decreased when pandan was planted between the areca nut forest in this study (Figures 2 and 7).

### 4.2. Regulatory Mechanisms of Planting Modes on Soil Microbial Diversity

Plant cultivation methods are the key factors affecting soil microbial communities and biological health [59]. The increase in crop varieties and the rational allocation of time and space between crops improve the soil rhizosphere microenvironment and nutrients, regulate the nutrient metabolism balance of microbes, and promote the functional potential as well as the relative stability of soil microbial communities [60]. The interactions among crop roots, rhizosphere soil, and soil microbes in the intercropping mode promote the accumulation of root exudates (i.e., organic acids) and the activity of soil catalytic substances (i.e., soil enzymes), increase the stability and anti-interference ability of the soil ecosystem, and improve microbial diversity [61]. Soil bacterial diversity was significantly increased after intercropping, which might be attributed to the regulation of soil properties and enzyme activities by soil bacteria (Figure 8). At the same time, more diverse plant litter and root secretions may have a positive impact on bacterial diversity [10,62].

SOC are a nutrient source for plants and soil microbes. The reduced SOC content reflects the decline in soil bacteria utilization of the carbon source and metabolic rate [63].

The negative correlation between SOC and the Ace and Chao indices indicated that the reduced content of SOC after intercropping was one of the main reasons for the addition in bacterial diversity. Phosphorus plays an important role in root development, stem growth, production of root secretions, and ATP synthesis [64]. The reduction in elemental phosphorus increased the complexity of the soil bacterial symbiotic network and affected the original metabolic level of soil bacteria [65]. Therefore, the soil bacterial diversity index was significantly and negatively correlated with both soil TP and SOP, thereby suggesting that soil bacteria were sensitive to the phosphorus content at the experimental site. Intercropping could significantly improve bacterial diversity by further reducing the phosphorus content in this study (Figures 1 and 5).

### 4.3. Key Regulatory Factors of Different Planting Modes on Soil Microbial Community Composition and Structure

The composition, structure, and function of the soil microbial community in the farmland ecosystem are closely related in the current study [66]. Reasonable intercropping is mainly performed indirectly by changing nutrient content and soil enzyme activities [67], which is beneficial to keep the soil microbial community structure stable, thereby improving the metabolic activity and functional diversity of beneficial microbes and, thus, inhibiting the growth of anaerobic bacteria, denitrifying bacteria, and other harmful microbes that occur in monoculture cultivation [68].

The decrease in Proteobacteria and Actinobacteria, the two abundant bacterial phyla, may be related to the biological properties of soil bacteria and different optimal growth environments. Proteobacteria perform the function of nitrogen fixation in the soil bacterial community and, using UE catalysis, convert soil's organic nitrogen to ammonia for plant uptake [69]. The decrease in Proteobacteria abundance after intercropping may be attributed to the significant reduction of soil TN content, because Proteobacteria, soil UE activity, and plants maintained the balance of soil nitrogen content in this study (Figure 8a). Actinobacteria genera such as *Actinomyces*, *Micromonospora*, and *Streptomyces* produce enzymes that dissolve phosphorus and accelerate the effective degradation of organic matter [70]. Thus, the decrease in soil TP content in the intercropping mode may have been one of the main reasons for the decrease in Actinobacteria-relative abundance in this study (Figures 1i and 6a) [71,72]. Acidobacteria are slow-growing oligotrophic bacteria with a K-selected life strategy, and Acidobacteria abundance is higher when the soil organic matter content is low [73]. The above conclusion was confirmed by the fact that the content of soil organic matter decreased, but Acidobacteria abundance increased in areca nut soil after the intercropping with pandan in this study. Species of Firmicutes are often found in nutrient-rich soil environments and can produce antimicrobial substances that promote plant growth and reduce the growth of pathogenic bacteria, while the acid soil environment may have a negative impact on Firmicutes abundance and activity [74]. The reduction of soil pH after intercropping may be the main reason for the decrease in Firmicutes in this study (Figures 4 and 5).

Soil bacteria and fungi responded differently to the modes of pandan intercropped with areca nut. In terms of fungal community composition, Ascomycota, Basidiomycota, and Rozellomycota were the three dominant fungal phyla, which were consistent with previous studies [75]. Compared with the significant changes in the bacterial community, the fungal community, except Ascomycota, was insensitive to changes in soil physicochemical properties, nutrients, and enzyme activities caused by intercropping. Ascomycota comprises decomposing fungi that decompose lignin and other organic substances that are not easily decomposed in soil, and it was also closely related to soil organic matter [76]. Thus, the decrease in the soil organic matter content might be the main reason for the decrease in Ascomycota abundance under the intercropping mode in this study (Figures 1 and 5).

### 4.4. Effects of Different Planting Modes on Soil Microbial Functional Groups

The FAPROTAX database was created to generate functional profiles by connecting individual organisms to ecologically relevant metabolic activities and applies to the functional annotation of bacteria associated with environmental samples [77]. The FAPROTAX prediction, which has been utilized frequently by other researchers, is arguably the best method for predicting probable microbial roles in samples [78]. Soil bacterial community function is highly correlated with the type of plants in the land, and changes in apoplectic inputs and root secretions in the intercropping system bring changes in the environment for soil bacteria to survive, which may lead to changes in soil bacterial community function [79]. Chemoheterotrophy and aerobic chemoheterotrophy have been found to be the most significant functions of the soil bacterial population in this study. Aerobic chemoheterotrophy can speed up the biodegradation of organic materials, and both are involved in the C cycle process. Chemoheterotrophic bacteria are decomposers in nature and are responsible for in situ restoration in all ecosystems [77]. The leaf litter of areca nut affects the growth of pandan, which needs to be cleared regularly. Therefore, there was a lack of carbon input from the areca nut litter in the intercropping model. During the experiment, the leaves of pandan were also harvested, so almost no litter material was produced, and the organic matter from litter material in the intercropping system was reduced, and the organic matter content decreased. In this study, the SOC, TN, and SAN contents in the soil were lower in the I mode than in the AM mode, and the closely related aerobic chemoheterotrophy and chemoheterotrophy functional communities were also significantly reduced, further demonstrating the close relationship between the bacterial functional communities and environmental factors (Figures 1 and 6).

For the functional determination of fungi, FUNGuild is an effective tool because it can identify the functional group roles of fungi from the perspective of trophic guilds, rather than from individual OTUs [80]. The results obtained from the FUNGuild procedure showed that soil saptrotrophs dominate the functions exercised by the fungi, which may play a central role in organic decomposition [51]. The proportion of functional groups of wood saprotroph fungi and lichenized fungi was increased under the intercropping mode, which related to the increased crop root biomass, but the specific functions still need to be further investigated in this study.

### 5. Conclusions

Intercropping pandan with areca nut had a positive impact on soil microbial diversity and dynamic balance, despite the fact that the bacterial community was more sensitive to the intercropping mode than the fungal community in the tropical plantations. We suggest that the decrease in soil nutrient content under the intercropping mode was the main reason for the increase in soil microbial diversity. Moreover, the change in soil enzyme activity may have changed the competitive relationships between the different kinds of microbes and nutrients, and then significantly changed the microbial community structure and functional groups. Complex interactions among soil properties, enzyme activity, and microbial communities not only resist the impact of intercropping management on soil functions but are also conducive to improving biodiversity in the tropical plantation.

**Author Contributions:** Writing—original draft preparation, Y.Z. (Yiming Zhong); conceptualization, X.Q.; methodology, A.Z.; software, Y.Z. (Yiming Zhong) and A.Z.; validation, J.T. and H.Y.; formal analysis, X.J.; investigation, J.T and Y.Z. (Yiming Zhong); resources, X.Q.; data curation, A.Z.; writing—review and editing, X.Q. and A.Z.; visualization, J.W. and A.Z.; supervision, S.H. and Y.Z. (Ying Zong); project administration, X.Q.; funding acquisition, H.Y. and X.J. All authors have read and agreed to the published version of the manuscript.

**Funding:** This research was supported by the Hainan Natural Science Foundation, China (No. 2019RC323). National Tropical Plants Germplasm Resource Center.

**Data Availability Statement:** Not applicable.

**Acknowledgments:** We thank Lihua Li, Shuangyan Qi, Jinshuang Li, Shaoguan Zhao and Jiang Zhong for their contributions to the preliminary work.

**Conflicts of Interest:** The authors declare no conflict of interest.

## Appendix A

**Table A1.** Soil dominant microbial composition under different planting patterns (phylum level).

|  | **PM** | **I** | **AM** |
|---|---|---|---|
| Bacteria |  |  |  |
| *Proteobacteria* | 19,202.33 ± 332.93 b | 20,230.33 ± 1974.88 b | 24,117.33 ± 866.37 a |
| *Actinobacteriota* | 16,801.00 ± 1021.31 b | 15,365.33 ± 712.89 b | 20,129.67 ± 298.15 a |
| *Acidobacteriota* | 10,725.33 ± 1697.53 a | 14,804.67 ± 2381.47 a | 13,546.33 ± 2433.41 a |
| *Firmicutes* | 14,715.33 ± 3308.49 a | 4676.67 ± 519.08 b | 6590.33 ± 1202.16 b |
| *Chloroflexi* | 6552.67 ± 992.70 ab | 5415.00 ± 93.15 b | 7745.33 ± 1054.78 a |
| *Bacteroidota* | 3069.67 ± 894.62 a | 3479.67 ± 1728.98 a | 4053.67 ± 167.99 a |
| *Myxococcota* | 2568.67 ± 160.48 a | 2770.00 ± 427.36 a | 2750.33 ± 315.78 a |
| *Methylomirabilota* | 458.33 ± 176.98 c | 2624.33 ± 531.17 a | 1812.00 ± 317.02 b |
| *Verrucomicrobiota* | 1187.67 ± 936.47 ab | 2081.67 ± 328.36 a | 706.33 ± 296.01 c |
| *Gemmatimonadota* | 920.33 ± 126.75 a | 1126.67 ± 29.30 a | 1074.33 ± 122.40 a |
| *Planctomycetota* | 915.67 ± 307.42 a | 1308.33 ± 496.08 a | 602.33 ± 101.11 a |
| *Bdellovibrionota* | 811.00 ± 149.08 a | 751.67 ± 255.58 a | 1102.67 ± 31.13 a |
| *unclassified Bacteria* | 420.00 ± 51.68 b | 909.67 ± 280.23 a | 994.33 ± 206.93 a |
| Fungi |  |  |  |
| *Ascomycota* | 107,293.00 ± 6816.14 a | 85,814.67 ± 10,179.97 b | 112,730.67 ± 5197.99 a |
| *Basidiomycota* | 14,402.00 ± 2356.95 a | 20,973.67 ± 18,341.30 a | 9764.33 ± 5609.52 a |
| *Unclassified Fungi* | 10,493.67 ± 3031.47 a | 13,498.33 ± 5525.81 a | 6377.67 ± 2092.01 a |
| *Rozellomycota* | 4314.00 ± 1480.56 a | 3867.67 ± 2074.86 a | 2791.00 ± 2524.00 a |

Note: Different letters indicate significant differences between treatments under the same soil microbes ($p < 0.05$).

**Table A2.** Relative abundance of bacterial functional groups based on intercropping under the FAPROTAX tool.

| Bacteria | PM | I | AM |
|---|---|---|---|
| Aerobic_chemoheterotrophy | 8183.83 ± 1148.68 a | 5707.67 ± 1096.84 b | 6216.83 ± 1358.33 b |
| Chemoheterotrophy | 7349.50 ± 3765.68 a | 5907.00 ± 1153.43 a | 6550.17 ± 1440.54 a |
| Animal_parasites_or_symbionts | 1519.00 ± 1102.83 a | 913.33 ± 291.12 a | 760.67 ± 135.74 a |
| Human_pathogens_all | 1490.5 ± 1089.56 a | 892.83 ± 301.78 a | 721.50 ± 146.22 a |
| Nitrate_reduction | 1326.00 ± 944.36 a | 735.67 ± 270.20 a | 677.83 ± 207.45 a |
| Nitrogen_fixation | 590.17 ± 316.09 b | 726.67 ± 91.09 ab | 897.17 ± 201.86 a |
| Predatory_or_exoparasitic | 477.67 ± 291.51 a | 504.67 ± 170.19 a | 500.17 ± 73.51 a |
| Human_pathogens_pneumonia | 538.50 ± 276.01 a | 538.33 ± 119.02 a | 398.67 ± 162.01 a |
| Invertebrate_parasites | 882.33 ± 887.74 a | 292.83 ± 211.82 a | 281.83 ± 157.54 a |
| Aromatic_compound_degradation | 561.50 ± 166.59 b | 401.33 ± 208.33 ab | 227.17 ± 104.81 a |
| Chitinolysis | 636.83 ± 657.41 a | 289.83 ± 315.06 a | 155.50 ± 148.64 a |
| Ureolysis | 316.00 ± 204.44 a | 337.00 ± 329.12 a | 326.83 ± 137.34 a |
| Phototrophy | 403.33 ± 428.91 a | 277.17 ± 69.48 a | 244.33 ± 51.35 a |
| Cellulolysis | 127.83 ± 71.19 b | 208.50 ± 147.37 ab | 503.67 ± 469.15 a |
| Photoautotrophy | 380.50 ± 428.29 a | 233.17 ± 66.01 a | 172.00 ± 85.81 a |
| Fermentation | 192.83 ± 104.82 a | 195.67 ± 165.37 a | 302.50 ± 152.66 a |
| Others | 2946.83 ± 1281.30 a | 2191.50 ± 614.30 a | 2000.17 ± 1076.32 a |

Note: Different letters indicate significant differences between treatments under the same bacterial functional groups ($p < 0.05$).

**Table A3.** Relative abundance of fungi functional groups based on intercropping under the FUN-Guild tool.

| Fungi | AM | I | PM |
|---|---|---|---|
| Wood_Saprotroph | 11,270.17 ± 7010.83 a | 4806.83 ± 1352.41 b | 7946.67 ± 1797.94 ab |
| Soil_Saprotroph | 858.17 ± 164.50 a | 417.33 ± 245.91 a | 894.17 ± 827.92 a |
| Plant_Pathogen | 1706.00 ± 565.32 a | 1601.50 ± 2539.11 a | 2047.33 ± 1870.07 a |
| Plant_Pathogen_Wood_Saprotroph | 1186.83 ± 891.84 a | 333.83 ± 267.63 b | 344.83 ± 608.14 b |
| Animal_Pathogen | 453.83 ± 361.55 ab | 689.67 ± 280.65 a | 221.83 ± 242.93 b |
| Endophyte | 16.17 ± 24.31 b | 645.67 ± 699.13 a | 433.17 ± 150.1 ab |
| Ectomycorrhizal | 28.17 ± 20.95 a | 696.00 ± 1028.55 a | 128.17 ± 100.88 a |
| Fungal_Parasite | 93.50 ± 54.34 a | 256.17 ± 315.36 a | 263.17 ± 198.15 a |
| Arbuscular_Mycorrhizal | 8.33 ± 5.47 b | 258.67 ± 239.75 a | 115.33 ± 112.58 ab |
| Epiphyte | 274.67 ± 423.17 a | 0.33 ± 0.82 a | 79.50 ± 193.27 a |
| Dung_Saprotroph | 101.00 ± 47.92 a | 27.00 ± 24.76 a | 134.83 ± 142.33 a |
| Plant_Saprotroph | 55.00 ± 27.40 a | 27.00 ± 21.72 a | 58.00 ± 58.36 a |
| Others | 4.17 ± 1.72 a | 21.67 ± 23.75 a | 36.67 ± 63.59 a |

Note: Different letters indicate significant differences between treatments under the same fungi functional groups ($p < 0.05$).

**Table A4.** Stepwise regression analysis model of soil dominant microorganisms and environmental factors.

| Soil Microorganisms | Regression Model | $R^2$ | F Value | p Value |
|---|---|---|---|---|
| *Actinobacteriota* | *Actinobacteriota* = 2831.888 + 11,364.551 × TP + 325.103 × TK | 0.951 | 57.843 | 0.000 |
| *Ascomycota* | *Ascomycota* = 10,079.723 + 5134.928 × SOC | 0.734 | 19.297 | 0.003 |
| *Chloroflexi* | *Chloroflexi* = −906.557 + 7427.543 × TN | 0.643 | 12.632 | 0.009 |
| *Firmicutes* | *Firmicutes* = −5205.866 + 388.736 × SAK | 0.846 | 38.479 | 0.000 |
| *Methylomirabilota* | *Methylomirabilota* = 5797.752 − 58.409 × EC | 0.908 | 69.196 | 0.000 |
| *Proteobacteria* | *Proteobacteria* = 5845.745 + 21.631 × URE | 0.761 | 22.343 | 0.002 |
| *Unclassified Bacteria* | *Unclassified Bacteria* = −8218.610 + 1235.298 × pH + 84.372 × SWC | 0.913 | 31.443 | 0.001 |
| *Verrucomicrobiota* | *Verrucomicrobiota* = 5945.663 − 258.262 × SOC | 0.575 | 9.469 | 0.018 |

**Table A5.** Redundancy analysis of soil bacteria and fungi, soil environmental variables.

| Name | Explains (%) | F | P |
|---|---|---|---|
| Environment-Bacteria | | | |
| pH | 24.5 | 13.6 | 0.002 |
| TP | 48.6 | 6.6 | 0.004 |
| POD | 11.2 | 3.6 | 0.022 |
| EC | 6.4 | 2.7 | 0.108 |
| UE | 3.8 | 2.0 | 0.146 |
| SOC | 2.3 | 1.4 | 0.312 |
| SAK | 1.8 | 1.3 | 0.434 |
| SOP | 1.4 | <0.1 | 1 |
| Environment-Fungi | | | |
| SAN | 23.5 | 2.2 | 0.144 |
| TK | 11.6 | 1.1 | 0.330 |
| BD | 28.0 | 3.8 | 0.102 |
| SWC | 8.2 | 1.2 | 0.354 |
| PPO | 18.5 | 5.5 | 0.064 |
| UE | 9.4 | 26.5 | 0.032 |
| POD | 0.3 | 0.7 | 0.554 |
| pH | 0.4 | <0.1 | 1 |

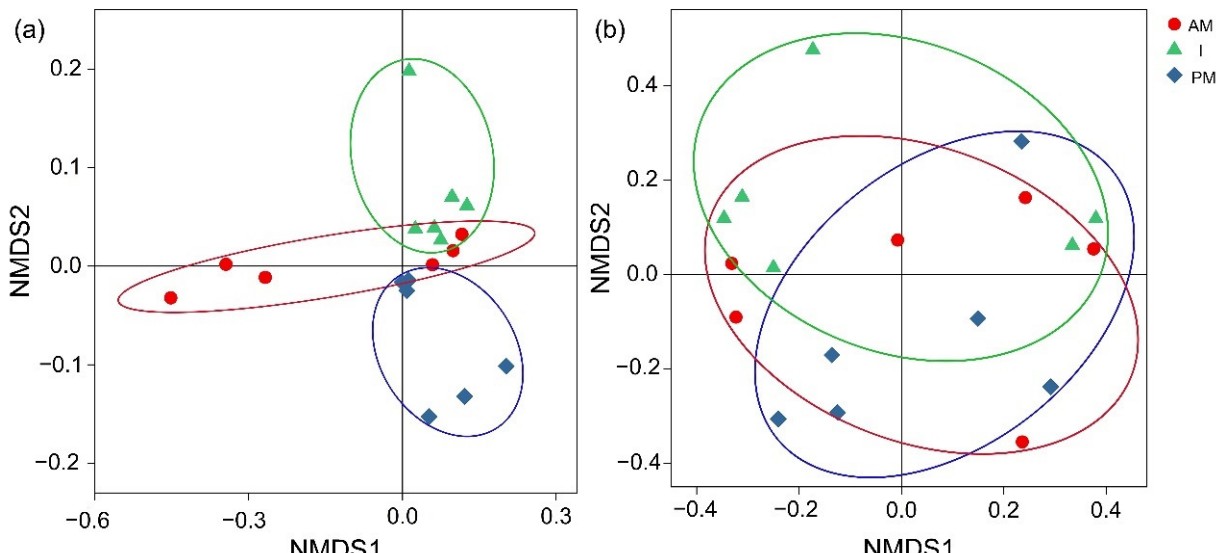

**Figure A1.** Effects of intercropping patterns on soil microbial (bacterial and fungal) beta diversity (NMDS) across the experimental period. AM represents areca nut monocropping; I represents areca nut intercropping with pandan; and PM represents pandan monocropping. ((**a**): $F = 1.9485$, $p = 0.09$; (**b**): $F = 1.969$, $p = 0.066$, calculated by PERMANOVA).

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
