# Peer review of "Effects of Intercropping Pandanus amaryllifolius on Soil Properties and Microbial Community Composition in Areca Catechu Plantations"

_forests, doi:10.3390/f13111814_

Round 1

Reviewer 1 Report

  • As stated in the introduction by the authors, intercropping allows two or more crops to be planted on the same field, which can optimize the utilization of sunlight, water, nutrient and space, and improve the overall production. In the meantime, intercropping may also influence soil properties and soil microbial community. Therefore, study on intercropping is meaningful in how to use the farmland efficiently to obtain the sustainable development of agriculture. My major concerns for the present manuscript are as follows: 1) it is not clear what are the questions to be addressed in this study. In the introduction section, the authors are expected to introduce what are already known and what are unknown in intercropping studies, thereby leading to the questions waiting to be resolved in the present study. While the authors simply stated in the last paragraph of introduction section that it is unclear whether the effects of long-term intercropping of cash crops on soil microbial communities are consistent with short-term intercropping of annual crops in tropical farmland. Indeed, I don’t see any comparison between short term intercropping and long-term intercropping; 2) I am not satisfied with the discussions or explanations on the results. For example, the authors argued that the increased microbial diversity was ascribed to the decreased SOM in the intercropping mode. I wonder why there is a decrease of SOM in the intercropping mode. The author simply attributed the decline in SOM to the increase in planting density, which is inconvincible and need a further explanation. As we all know, SOM content is very important in agriculture. A decrease of SOM indicates that the farming mode is not suitable for sustainable agriculture. I strongly recommend that the authors focus on the variation of SOM, rather than the correlation between soil properties and soil microbial community, which I don’t think is meaningful and important.

Minor concerns:

L38, why is soil health in the keywords?

L58-73, this paragraph mainly explains the definition of enzymes, which are actually well known and can be simplified.

L75-78, these two sentences convey the similar meaning, which can be combined into one sentence.

L197, “P<0.001” should be “P<0.001”, the same below.

L332, “three” instead of “several” planting modes

L383-384, inconvincible explanation. Increase in planting density does not necessarily mean the loss of N in soil.

L401, promote the improvement, it is not necessary to use these two words together.

L404, I don’t think it is correct to state “SOC improves soil structure…”

L420, inappropriate expression, please reword.

L422, what is “balance of microbes”?

L430, to supply plant?

L438, can the organic matter content be absent?

L482, what does “intermediate” mean here?

Author Response

Response to Reviewer 1 Comments

Point 1:- It is not clear what are the questions to be addressed in this study. In the introduction section, the authors are expected to introduce what are already known and what are unknown in intercropping studies, thereby leading to the questions waiting to be resolved in the present study. While the authors simply stated in the last paragraph of introduction section that it is unclear whether the effects of long-term intercropping of cash crops on soil microbial communities are consistent with short-term intercropping of annual crops in tropical farmland. Indeed, I don’t see any comparison between short term intercropping and long-term intercropping.

Response 1: Thank you for your comments and ideas. Tropical cash crops are generally perennial crops, however, most of the current research has focused on food crops such as rice and maize, and it is not known whether tropical cash crops have the same effect on soil microorganisms as food crops in the early stages of cultivation, which is the scientific issue discussed in this manuscript.

Point 2:- I am not satisfied with the discussions or explanations on the results. For example, the authors argued that the increased microbial diversity was ascribed to the decreased SOM in the intercropping mode. I wonder why there is a decrease of SOM in the intercropping mode. The author simply attributed the decline in SOM to the increase in planting density, which is inconvincible and need a further explanation. As we all know, SOM content is very important in agriculture. A decrease of SOM indicates that the farming mode is not suitable for sustainable agriculture. I strongly recommend that the authors focus on the variation of SOM, rather than the correlation between soil properties and soil microbial community, which I don’t think is meaningful and important.

Response 2: We have also enhanced our manuscript to explore the mechanism of the effect of organic matter changes on soil microorganisms. We believe that the decline in soil organic matter in this intercropping model may be related to two reasons.

  1. The leaf litters matter of areca nut affects the growth of pandan, which needs to be cleared regularly. Therefore, the intercropping mode lacks carbon input from areca nut foliage, and the intercropping pattern also lacks carbon input from areca nut foliage because it is an evergreen herb and its harvesting site is the leaves.
  2. The rapid growth of pandan converts more photosynthetic products into aboveground biomass such as stolon and leave, with less transfer to the ground, and the shallow distribution of the pandan root system competes with the areca nut root system, causing the areca nut root system to grow deeper, reducing the carbon input from the root system in the surface soil.

We speculate that there may be a significant increase of plant roots (biochar carbon pool) in the soil, and the next work will strengthen the monitoring of plant roots.

Minor concerns:

Point 1:-L38, why is soil health in the keywords?

Response 1: Thank you for your ideas. The manuscript focuses on the effect of intercropping on soil microbial community structure, and the relative stability of soil microbial communities is an important indicator of soil health, so we wanted to include the keyword soil health in the manuscript. On page 2, Lines 41.

Point 2:-L58-73, this paragraph mainly explains the definition of enzymes, which are actually well known and can be simplified.

Response 2: Thank you for your comments, we have combined and abbreviated the introduction of soil enzyme activity with the previous paragraph. “Common soil enzymes, such as catalase (CAT), acid phosphatase (ACP), urease (UE), and invertase, play a catalytic role in the decomposition of plant and animal residues, accelerating their biochemical reactions [15]. UE participates in the ammoniation of organic nitrogen in the nitrogen cycle in the farmland ecosystem to produce plant available nitrogen [17]. UE activity determines the transfer rate of soil nutrients [18]. Peroxidase (POD) degrades lignin and coupled polysaccharides and is related to the degradation of polyphenols produced by soil fungi [19,20]. There is a close relationship between soil enzyme activities and soil microbial characteristics because soil microbes are capable of secreting a range of enzymes, and changes in soil enzyme activities reflect changes in nutrient requirements and metabolic activity of soil microbes [21]. Intercropping affects the relationship between soil enzyme activity and microbial community by changing soil properties and microenvironments and then regulates the structure and function of the soil microbial community [22].” On page 2, Lines 60-71.

Point 3:-L75-78, these two sentences convey the similar meaning, which can be combined into one sentence.

Response 3: Thanks to your comments, we have merged the two sentences.We changed ”Microbial community composition is related to soil functions and ecosystem sustainability due to its involvement in soil organic matter dynamics and nutrient cycling pro-cesses. Microbial abundance and diversity play a key role in maintaining soil func-tion and quality as well as metabolic activity.” to ” Microbial community composition is related to soil function and ecosystem sustainability because it is involved in soil organic matter dynamics and nutrient cycling processes as well as in the metabolism of the soil system.” On page 2, Lines 78-80.

Point 4:-L197, “P<0.001” should be “P<0.001”, the same below.

Response 4: Sorry for our error in formatting. We have made correction according to the Reviewer’s comments. We have checked the relevant font issues. On page 5, Lines 228-237.

Point 5:-L332, “three” instead of “several” planting modes

Response 5: We have made correction according to the Reviewer’s comments. On page 14, Lines 382.

Point 6:-L383-384, inconvincible explanation. Increase in planting density does not necessarily mean the loss of N in soil.

Response 6: Thank you for your comments, we have corrected and rephrased the paragraph.” The increase in planting density may intensify crop competition for nitrogen (N), or it may be that pandan has a stronger demand for N in the soil than betel nut, resulting in a decline in soil N content in the intercropping pattern. This study confirmed the viewpoint above since the intercropping pattern's soil alkaline N content was closer to that of the monocropping model [48] (Figure2, h).” On page 16, Lines 443-447.

Point 7:-L401, promote the improvement, it is not necessary to use these two words together.

Response 7: We have made correction according to the Reviewer’s comments. On page 16, Lines 462.

Point 8:-L404, I don’t think it is correct to state “SOC improves soil structure…”

Response 8: We have made correction according to the Reviewer’s comments. We changed the sentence in the manuscript. “SOC as a nutrient source for plants and soil microbes.” On page 16, Lines 467-469.

Point 9:-L420, inappropriate expression, please reword.

Response 9: We have made correction according to the Reviewer’s comments. We have rephrased the sentences in the manuscript. “Reasonable intercropping is mainly performed indirectly by changing nutrient content and soil enzyme activities [60], which is beneficial to keep the soil microbial community structure stable, improving the metabolic activity and functional diversity of beneficial microbes and thus inhibiting the growth of anaerobic bacteria, denitrifying bacteria, and other harmful microbes that occur in monoculture cultivation [61]. On page 17, Lines 484-487.

Point 10:-L422, what is “balance of microbes”?

Response 10: Thank you for your correction, we have made changes to your comments.” which is beneficial to keep the soil microbial community structure stable.” On page 17, Lines 486-487.

Point 11:-L430, to supply plant?

Response 11: We apologize for the typographical errors and mistakes made in the manuscript. Thank you for your questions, we have made changes to the manuscript. “Using UE catalysis, convert soil's organic nitrogen to ammonia for plant uptake [62].” On page 17, Lines 493-495.

Point 12:-L438, can the organic matter content be absent?

Response 12: We apologize for the typographical errors and mistakes made in the manuscript. Thank you for your questions, we have made changes to the manuscript. “Thus, the decrease of soil TP content in intercropping mode may have been become one of the main reasons for the decrease ininhibition of Actinobacteria relative abundance in this study (Figure 1i, 7a, 9a) [64,65].” On page 17, Lines 504.

Point 13:-L482, what does “intermediate” mean here?

Response 13: We apologize for the typographical errors and mistakes made in the manuscript. Thank you for your questions, we have made changes to the manuscript. “The proportion of functional groups of Wood Saprotroph fungi and Lichenized fungi was increased under the intermediate cropintercropping mode , which may be related to the increased crop root biomass, but the specific functions still need to be further investigated in this study.” On page 18, Lines 554.

Reviewer 2 Report

Zhong et al. studied soil properties, enyzme activities, and microbial communities in three different treatments - two monocultures and an intercropping system. The premise and methods are sound, although there are a few improvements that could be made in the statistical analysis and figures. I am confident that the authors will be able to revise their manuscript and make it satisfactory for publication in Forests. I have extensively edited the English grammar thoughout, and made some other comments (see below). I also suggest that the authors think critically about causality and tone down some of their language.

Title: Should say either “in an Areca catechu Plantation” or “in Areca catechu Plantations”

Abstract:

Line 23: change “microbial” to “bacterial and fungal”

Line 23: Correlation analysis and redundancy analysis

Line 25: delete “were studied”

Line 30: phyla

Line 31: delete “were”

You didn’t say anything about the FAPROTAX or FUNguild results in the abstract. Perhaps you could include that?

It’s unclear what you mean by “function realization” and this is awkward wording.

Introduction:

Line 42: deleted “succession and”. Is succession a problem? I think it’s better just to say degradation.

Line 46: change “improve” to “improving”

Line 48: change “considerable yield advantages” to “considerably increasing yield”

Line 48: change “stimulated” to “stimulating”

Line 49: change “maintain” to “and maintaining”

Line 52: change “as” to “is”

Line 55: change “microbe” to “microbes”

Line 73: microenvironments

Line 77: extra space

Line 79: it is important to recognize the contributions of archaea and protists though too…

Line 90: somewhere in this section, it’s probably worth mentioning legumes and N fixation as important considerations/processes

Line 94: change “The previous study” to “Previous studies”

Line 101: change “; for example,” to “including production of”

Line 110: and microbial

Line 110: explore the key mechanism of how the intercropping system alters soil microbial community diversity and structure

Line 112: functional change on soil health in a 

Paragraphs 2,3,4 of the intro sometimes seem to repeat similar points. Check this and make sure the organization is clear.

Methods:

Line 128: Each block was replicated 6 times.

Line 133: The authors state that water/fertilizer/pest management was the same, but they need to provide more details about each of these three factors, even if they were the same, as this information is crucial to understand the soil and microbial response.

Line 154: describe what you mean by “using kits”. Also, I assume this was “potential activity” of the enzymes measured after an incubation. This needs to be clarified for readers and more details provided.

Line 166: The data were analyzed

Line 169: Software programs need to be cited in this paragraph and elsewhere (e.g. FAPROTAX, FUNguild etc).

Line 172: Why did the authors choose OTUs instead of amplicon sequence variants (ASVs)?

Line 173: Need to state which version of the SILVA database was used.

Line 174: standard for normalization…this is vague, how exactly were the data normalized?

Line 174: should state here what the mean (SE) sequencing depth per sample was.

Line 183: How many archaeal reads were there? Were these removed?

Line 185: Should state what the dependent variables are - enzymes, nutrients, microbes etc.

What about the starting soils? Are there soil microbe data from 2015 before the experiment started? That would be helpful as a control.

Results:

Line 196: Compared to the AM mode

Line 204: Compared to the AM mode

Line 208: than in the PM mode

Line 209: between the AM and PM modes in the 

Line 210: period after activity and start new sentence.

Line 211: were altered

Line 221: Figure 1 caption: Are these really “correlations”? Change “correlation” to “comparison”? I think this should also be ANOVA and Tukey posthoc.

Figure 2: Authors should use the same colors as in Figure 1, or vice versa.

Line 239: delete “ compared with the a-diversity index”?

Lines 244-247: You must run a PERMANOVA (and add this to the methods too) to be able to make a statement like this, and report the p, pseudo-F, and R2 values. NMDS is purely for visualization, you must run PERMANOVA to make statements about differences or lack of differences. PERMDISP should be run as well. I usually run these with the “adonis2” and “betadisper” functions in the vegan R package.

Figure 3: Need to clarify that (I assume) the first 4 panels are bacteria and the second 4 are fungi.

Lines 251-251: “represents” instead of “represent”. Same in Figure 4 caption.

Figure 4: again, confirm that bacterial is a and fungal is b

Figure 4: it would be great to show environmental vectors on this ordination (e.g., envfit in the vegan R package), this would help visualize the environmental context of the microbial differences. I suggest combining Figures 4 and 9 instead of having a separate figure 9.

Figure 5: this figure needs to be remade to show the variation in each treatment, i.e., show 6 bars for each treatment (1 for every sample)

Figure 5: also these are clearly not percentages so the caption needs to be updated to accurately describe the abundances

Line 273: period after (absolute difference) and start new sentence. Also it would be helpful to decide to present either relative or absolute differences but don’t switch between the two. Just choose one and report that.

Line 273: change “tends to increase” to “increased”

Figure 6: Ascomycota is cut off, please fix

Line 283: Change “groups” to “phyla”

Line 284: clarify again here what the correlation cutoff is (p value and/or r value) to be shown here.

Line 291: chemoheterotrophic bacteria

Line 293: chemoisomeric bacteria in the AM and I modes was significantly lower than in the PM

Line 294: delete “However,”. And I think you mean “The main functional prediction…”

Line 297: the PM treatment

Line 298: the AM and I treatments

Line 299: under the intercropping treatment

Line 300: than that in the AM or PM treatments

Figure 7. Need to add more to the methods about how you managed the FAPROTAX and FUNguild assignments. For example, in FUNguild, which confidence levels did you accept? Also what did you do with taxa that were assigned to multiple guilds? In the caption, change “represent” to “represents”

Line 320: was

Figure 9: I would delete this and just show vectors on Figure 4. If you want to assess relationships between environmental variables and phyla abundances, you can just run correlations instead of showing it like this. Meanwhile, a distance-based redundancy analysis would greatly help the interpretation of the microbial composition and be a more informative Figure 4.

Discussion:

Line 368: what do you mean general researchers? Do you mean “In general, researchers…”

Line 370: systems

Line 370: change A to a and delete “research”

Line 403: I would also attribute the increase in diversity to the present of more diverse plant litter and exudates

Line 409: this is not correct - you said in line 402 that intercropping increased bacterial diversity (which is what Figure 3 shows)

Line 422: Mouhamadou ref needs to be formatted as the others with a number.

Line 435: change “become” to “may have been”

Line 436: change “inhibition of” to “decrease in”

Line 437: Acidobacteria are

Line 438: change “absent” to “low”

Line 456: but why wouldn’t the decrease in SOM also cause a decline in Ascomycota?

Line 461: change “now” to “arguably” . The makers of PiCRUST and PiCRUST2 would disagree…

Line 466: Chemoheterotrophy (change to one word, here and elsewhere)

Line 469: Chemoheterotrophic bacteria

Line 471: chemoheterotrophic groups

Line 472: change “chemical heterotrophic” to “chemoheterotrophic”

Line 473: how do you know there was increased fine root turnover? You did not measure this.

Line 480: soil saptrotrophs dominate

Line 482: change “intermediate crop” to “intercropping mode, which”

Conclusions: 

In general, need to tone down the language some, here and in the discussion.

Line 487: despite the fact that

Line 487: the intercropping mode

Line 488: We suggest the decrease…( you are suggesting this, you do not know this for sure)

Line 490: may have changed the competition (again, it may have done this, but you don’t know for sure)

Line 490: change “competition relationship” to “competitive relationships”

Line 492: changed (past tense)

What about plant growth? This is a key piece missing from this study. It would be great to connect all of this soil work back to the important outcome variable, which is the crop production!

Some other things to consider:

It seems like the authors need to clarify that the enzymes are produced by microbes. Sometimes the authors imply that enzyme activity is affecting microbial communities but it is likely the other way around. Probably the crop management system changes soil properties and microbes and the enzyme activities are in turn affected by this.

The authors should comment on the 3 points in Figure 4a that are out to the left (low axis 1 values). Why are these 3 replicates different than the other 3 replicates? It is cool that 3 of the replicates in the I mode fall in between the AM and PM modes in ordination space!

Author Response

Response to Reviewer 2 Comments

Title:

Point1: -Should say either “in an Areca catechu Plantation” or “in Areca catechu Plantations”

Response 1: We have made correction according to the Reviewer’s comments.From “Areca catechu Plantation” to “Areca catechu Plantations”. In line 2.

Abstract:

Point2: -Line 23: change “microbial” to “bacterial and fungal”

Response 2: We have made correction according to the Reviewer’s comments. ” were used to analyze and predict the bacteria and fungi microbial community functions, respec-tively.” In line 23.

Point3: -Line 23: Correlation analysis and redundancy analysis

Response 3: We have made correction according to the Reviewer’s comments.” Correlation analysis and redundancy analysis was used”. In line 23-24.

Point4: -Line 25: delete “were studied”

Response 4: We have made correction according to the Reviewer’s comments. In line 25.

Point5: -Line 30: phyla

Response 5: We have made correction according to the Reviewer’s comments.” The dominant bacterial and fungal phyla”. In line 33.

Point6: -Line 31: delete “were”

Response 6: We have made correction according to the Reviewer’s comments. In line 34.

Point7: -You didn’t say anything about the FAPROTAX or FUNguild results in the abstract. Perhaps you could include that?

Response 7: Thank you for your comments, we have added relevant content to the summary.” Functional predictions of fungal microbial communities by FAPROTAX and Funguild indicated that chemoheterotrophy, aerobic chemoheterotrophy and soil saprotroph were the most dominant functional communities.” In line 27-29.

Point8: -It’s unclear what you mean by “function realization” and this is awkward wording.

Response 8: We have made correction according to the Reviewer’s comments. We removed the statement “and function realization”. In line 39.

Introduction:

Point9: -Line 42: deleted “succession and”. Is succession a problem? I think it’s better just to say degradation.

Response 9: We have made correction according to the Reviewer’s comments. We have removed the description of “succession and”, it's our negligence to get this wrong description. In line 45.

Point10: -Line 46: change “improve” to “improving”

Response 10: We have made correction according to the Reviewer’s comments. We have changed “improve” to “improving”. In line 49-50.

Point11: -Line 48: change “considerable yield advantages” to “considerably increasing yield”

Response 11: We have made correction according to the Reviewer’s comments. Thank you for improving the accuracy of the description. We have changed “considerable yield advantages” to “considerably increasing yield”. In line 51.

Point12: -Line 48: change “stimulated” to “stimulating”

Response 12: We have made correction according to the Reviewer’s comments. We have changed ”stimulated” to “stimulating”. In line 52.

Point13: -Line 49: change “maintain” to “and maintaining”

Response 13: We have made correction according to the Reviewer’s comments. We have changed “maintain” to “and maintaining”. In line 53.

Point14: -Line 52: change “as” to “is”

Response 14: We have made correction according to the Reviewer’s comments. We have changed “as” to “is”. In line 56.

Point15: -Line 55: change “microbe” to “microbes”

Response 15: We have made correction according to the Reviewer’s comments. We have changed “microbe” to “microbes”. In line 59.

Point16: -Line 73: microenvironments

Response 16: We have made correction according to the Reviewer’s comments. We have changed “microenvironment” to “microenvironments”. In line 98.

Point17: -Line 77: extra space

Response 17: We have made correction according to the Reviewer’s comments. We have removed the extra spaces. In line 80.

Point18: -Line 79: it is important to recognize the contributions of archaea and protists though too…

Response 18: We have made correction according to the Reviewer’s comments. In line 84-85.

Point19: -Line 90: somewhere in this section, it’s probably worth mentioning legumes and N fixation as important considerations/processes

Response 19: Thank you for your comments. We have made correction according to the Reviewer’s comments. The processes and causes of nitrogen-related changes have been explored in the discussion of other intercropping models. We have added the relevant description at the end of this paragraph. ” Legumes and nitrogen fixation may increase the nitrogen content of the soil, but in other intercropping systems, different effects may occur.” In line 104-106.

Point20: -Line 94: change “The previous study” to “Previous studies”

Response 20: We have made correction according to the Reviewer’s comments. We have changed “The previous study” to “Previous studies”. In line 109.

Point21: -Line 101: change “; for example,” to “including production of”

Response 21: We have made correction according to the Reviewer’s comments. We have changed “; for example,” to “including production of”. In line 116.

Point22: -Line 110: and microbial

Response 22: We have made correction according to the Reviewer’s comments. We have changed “microbial” to “and microbial”. In line 126.

Point23: -Line 110: explore the key mechanism of how the intercropping system alters soil microbial community diversity and structure

Response 23: We have made correction according to the Reviewer’s comments. We have rewritten the phrase “explore the key mechanism of how the intercropping system alters soil microbial community diversity and structure”. In line 126-127.

Point24: -Line 112: functional change on soil health in a 

Response 24: We have made correction according to the Reviewer’s comments. We have rewritten the phrase “investigate the effect of soil microbial community functional change on soil health in tropical intercropping system.” In line 130.

Point25: -Paragraphs 2,3,4 of the intro sometimes seem to repeat similar points. Check this and make sure the organization is clear.

Response 25: We have made correction according to the Reviewer’s comments. We have adjusted the structure and description of the introduction part. In line 44-131.

Methods:

Point26: -Line 128: Each block was replicated 6 times.

Response 26: We have made correction according to the Reviewer’s comments. We have changed “Each block had three plots” to “Each block was replicated 6 times”. In line 144.

Point27: -Line 133: The authors state that water/fertilizer/pest management was the same, but they need to provide more details about each of these three factors, even if they were the same, as this information is crucial to understand the soil and microbial response.

Response 27: Sorry to confuse the reviewers, in this experiment we chose the same time to plant betel nut in 2019, watered for one hour every three days at 5 pm by using water and fertilizer integration facilities (same amount of watering), sprayed in equal amounts using the same brand size of water soluble fertilizer, and did not use pesticides because of the low number of pests and diseases in betel nut and Xiangluodu in this experiment. In line .

Point28: -Line 154: describe what you mean by “using kits”. Also, I assume this was “potential activity” of the enzymes measured after an incubation. This needs to be clarified for readers and more details provided.

Response 28: We have made correction according to the Reviewer’s comments. Thank you for your comments. We measured a total of five soil enzyme activities, and the principles are explained in the manuscript. We are sorry that some of the detailed reagents could not be determined due to the use of kits for enzyme activity measurement. In line 173-179.

Point29: -Line 166: The data were analyzed

Response 29: We have made correction according to the Reviewer’s comments. We have changed “The data was analyzed” to “The data were analyzed”. In line 192.

Point30: -Line 169: Software programs need to be cited in this paragraph and elsewhere (e.g. FAPROTAX, FUNguild etc).

Response 30: We have made correction according to the Reviewer’s comments. We have cited the software used in the manuscript in section 2.7 analysis methods, please check. In line 211.

Point31: -Line 172: Why did the authors choose OTUs instead of amplicon sequence variants (ASVs)?

Response 31: Thank you for your question. OTU is still used in most articles and we refer to the following papers:

[1] Yang, J.; Duan, Y.; Liu, X.; Sun, M.; Wang, Y.; Liu, M.; Zhu, Z.; Shen, Z.; Gao, W.; Wang, B.; Chang, C.; Li, R. Reduction of banana fusarium wilt associated with soil microbiome reconstruction through green manure intercropping. Agriculture, Ecosystems & Environment 2022, 337, 108065.

[2] Bian, F.; Zhong, Z.; Li, C.; Zhang, X.; Gu, L.; Huang, Z.; Gai, X.; Huang, Z. Intercropping improves heavy metal phytoremediation efficiency through changing properties of rhizosphere soil in bamboo plantation. J Hazard Mater 2021, 416, 125898.

[3] Wang, J.; Lu, X.; Zhang, J.; Wei, H.; Li, M.; Lan, N.; Luo, H. Intercropping perennial aquatic plants with rice improved paddy field soil microbial biomass, biomass carbon and biomass nitrogen to facilitate soil sustainability. Soil and Tillage Research 2021, 208, 104908.

Point32: -Line 173: Need to state which version of the SILVA database was used.

Response 32: We have made correction according to the Reviewer’s comments. We added this section “rRNA SILVA database (v 138) and Unite 8.0 fungi database”. In line 200.

Point33: -Line 174: standard for normalization…this is vague, how exactly were the data normalized?

Response 33: Thank you for your question. We apologize for not writing the specific process of standardization in the manuscript due to our oversight, we use the method of draw level subsample for standardization: the series of all samples are randomly selected to that amount of data according to the minimum number of sample series, and then subsequent analysis is performed. Additional explanation has been provided in the manuscript.

Point34: -Line 174: should state here what the mean (SE) sequencing depth per sample was.

Response 34: Thank you for your comments. For microbial diversity there is only the concept of sequencing volume, not the concept of sequencing depth, and there is something related to sequencing volume that is described in the results section. We looked up the original data and added schedules A6 and A7 to the end of the manuscript.

Point35: -Line 183: How many archaeal reads were there? Were these removed?

Response 35: Thank you for your question. This test was only performed for bacteria and fungi, not for archaea, so there is no data related to archaea readers.

Point36: -Line 185: Should state what the dependent variables are - enzymes, nutrients, microbes etc.

Response 36: We have made correction according to the Reviewer’s comments. We have added the relevant content.” The experimental indicator (soil physical and chemical properties, soil enzyme activity, soil fungal bacterial diversity and community structure, prediction of soil fungal bac-terial functional communities) was analyzed by one-way ANOVA to determine dif-ferences between intercropping and monocropping modes.” In line 213-215.

Point37: -What about the starting soils? Are there soil microbe data from 2015 before the experiment started? That would be helpful as a control.

Response 37: Thank you for your questions. Initial indicators were not available at the beginning of the trial establishment, and this paper only needs to compare the differences between different cropping patterns, which is our regret.

Results:

Point38: -Line 196: Compared to the AM mode

Response 38: We have made correction according to the Reviewer’s comments. We have changed “Compared with the AM mode” to “Compared to the AM mode”. In line 227.

Point39: -Line 204: Compared to the AM mode

Response 39: We have made correction according to the Reviewer’s comments. We have changed “Compared with the AM mode” to “Compared to the AM mode”. In line 236.

Point40: -Line 208: than in the PM mode

Response 40: We have made correction according to the Reviewer’s comments. We have changed “than the PM mode” to “than in the PM mode”. In line 240.

Point41: -Line 209: between the AM and PM modes in the 

Response 41: We have made correction according to the Reviewer’s comments. We have changed “between the AM and PM modes on the” to “between the AM and PM modes in the”. In line 241.

Point42: -Line 210: period after activity and start new sentence.

Response 42: We have made correction according to the Reviewer’s comments. In line 242.

Point43: -Line 211: were altered

Response 43: We have made correction according to the Reviewer’s comments. We have changed “was” to “were”. In line 243.

Point44: -Line 221: Figure 1 caption: Are these really “correlations”? Change “correlation” to “comparison”? I think this should also be ANOVA and Tukey posthoc.

Response 44: We have made correction according to the Reviewer’s comments. Thanks for the reminder, we have corrected the statement in the note. “Soil properties under different cropping patterns (n=9). Different letters indicate significant dif-ferences (ANOVA, P < 0.05, Tukey’s HSD post hoc analysis) among different intercropping patterns. *Correlation is significant at the 0.05 level; ** Correlation is significant at the 0.01 level; *** Corre-lation is significant at the 0.001 level. AM represents areca nut monocropping; I represents arcea nut intercropping with pandan; PM represents pandan monocropping.” In line 253-257.

Point45: -Figure 2: Authors should use the same colors as in Figure 1, or vice versa.

Response 45: We have made correction according to the Reviewer’s comments. We have standardized the color scheme of the images in the manuscript. In figure 1-3..

Point46: -Line 239: delete “ compared with the a-diversity index”?

Response 46: We have made correction according to the Reviewer’s comments. We have deleted “ compared with the a-diversity index”. In line 273-274.

Point47&51&68: Dear reviewers, all three questions are related to Figure 4, so we put them together in our response.

Point47: -Lines 244-247: You must run a PERMANOVA (and add this to the methods too) to be able to make a statement like this, and report the p, pseudo-F, and R2 values. NMDS is purely for visualization, you must run PERMANOVA to make statements about differences or lack of differences. PERMDISP should be run as well. I usually run these with the “adonis2” and “betadisper” functions in the vegan R package.

Point51: -Figure 4: it would be great to show environmental vectors on this ordination (e.g., envfit in the vegan R package), this would help visualize the environmental context of the microbial differences. I suggest combining Figures 4 and 9 instead of having a separate figure 9.

Point66: -Figure 7. Need to add more to the methods about how you managed the FAPROTAX and FUNguild assignments. For example, in FUNguild, which confidence levels did you accept? Also what did you do with taxa that were assigned to multiple guilds? In the caption, change “represent” to “represents”

Point68: -Figure 9: I would delete this and just show vectors on Figure 4. If you want to assess relationships between environmental variables and phyla abundances, you can just run correlations instead of showing it like this. Meanwhile, a distance-based redundancy analysis would greatly help the interpretation of the microbial composition and be a more informative Figure 4.

Response 47&51&66:

Thank you for your comments and detailed knowledge of the R software. We have moved the NMDS and PERMANOVA results from the manuscript to the attached table for your review.

In the study, RDA is more used to analyze the interrelationship between environmental factors and species. We have used RDA analysis for analyses concerning the relationship between soil environmental variables and soil fungi and bacteria to replace the previous NMDS analysis. And based on your suggestion, we have added the ranking of samples in the RDA analysis and refer to the following academic papers.

Shao, P.; Liang, C.; Rubert-Nason, K.; Li, X.; Xie, H.; Bao, X. Secondary successional forests undergo tightly-coupled changes in soil microbial community structure and soil organic matter. Soil Biology and Biochemistry 2019, 128, 56-65.

Shuai Liu, Zhanyu Wang, Junfeng Niu, Kaikai Dang, Shuke Zhang, Shiqiang Wang, Zhezhi Wang. Changes in physicochemical properties, enzymatic activities, and the microbial community of soil significantly influence the continuous cropping of Panax quinquefolius L. (American ginseng). Plant Soil. 2021, 463, 427-446.

Point48: -Figure 3: Need to clarify that (I assume) the first 4 panels are bacteria and the second 4 are fungi.

Response 48: We have made correction according to the Reviewer’s comments. We have confirmed and added the relevant comments. In line 284-285.

Point49: -Lines 251-251: “represents” instead of “represent”. Same in Figure 4 caption.

Response 49: We have made correction according to the Reviewer’s comments. We have changed “represents” to “represent”. In line 285-286.

Point50: -Figure 4: again, confirm that bacterial is a and fungal is b

Response 50: We have made correction according to the Reviewer’s comments. We have confirmed and added the relevant comments. In line 292-293.

Point52: -Figure 5: this figure needs to be remade to show the variation in each treatment, i.e., show 6 bars for each treatment (1 for every sample)

Response 52: We have made correction according to the Reviewer’s comments. We have redrawn the figure 5.

Point53: -Figure 5: also these are clearly not percentages so the caption needs to be updated to accurately describe the abundances

Response 53: We have made correction according to the Reviewer’s comments. After redrawing figure 5, the new image indicates the percentage of bacteria and fungi, please check. In line 310.

Point54: -Line 273: period after (absolute difference) and start new sentence. Also it would be helpful to decide to present either relative or absolute differences but don’t switch between the two. Just choose one and report that.

Response 54: Thank you for pointing out that we used relative differences in the manuscript to describe the differences between the different cropping patterns (we have standardized to absolute differences in this section to avoid confusing the reader). In line 322-329.

Point55: -Line 273: change “tends to increase” to “increased”

Response 55: We have made correction according to the Reviewer’s comments.We have changed “tends to increase” to “increased”. In line 325.

Point56: -Figure 6: Ascomycota is cut off, please fix

Response 56: We have made correction according to the Reviewer’s comments. We have repositioned the fungal names in this figure, please check.

Point57: -Line 283: Change “groups” to “phyla”

Response 57: We have made correction according to the Reviewer’s comments. We have changed “groups” to “phyla”. In line 337.

Point58: -Line 284: clarify again here what the correlation cutoff is (p value and/or r value) to be shown here.

Response 58: We have made correction according to the Reviewer’s comments. We have confirmed and added the relevant comments. In line 342.

Point59: -Line 291: chemoheterotrophic bacteria

Response 59: We have made correction according to the Reviewer’s comments. We have changed

“chemo heterotrophy” to “chemoheterotrophy”. And to correct other places in the article where it appears, please check. In line 345-348.

Point60: -Line 293: chemoisomeric bacteria in the AM and I modes was significantly lower than in the PM

Response 60: We have made correction according to the Reviewer’s comments. We have changed “chemo isomeric bacteria in AM and I mode was significantly lower than in PM treatment in this study” to “chemoisomeric bacteria in the AM and I modes was significantly lower than in the PM”. In line 348-350.

Point61: -Line 294: delete “However,”. And I think you mean “The main functional prediction…”

Response 61: We have made correction according to the Reviewer’s comments. We have changed “However, the functional prediction” to “The main functional prediction”. In line 351.

Point62: -Line 297: the PM treatment

Response 62: We have made correction according to the Reviewer’s comments. We have changed “PM treatment” to “the PM treatment”. In line 354-355.

Point63: -Line 298: the AM and I treatments

Response 63: We have made correction according to the Reviewer’s comments. We have changed “AM and I treatments” to “the AM and I treatments”. In line 355-356.

Point64: -Line 299: under the intercropping treatment

Response 64: We have made correction according to the Reviewer’s comments. We have changed “under intercropping treatment” to “under the intercropping treatment”. In line 357.

Point65: -Line 300: than that in the AM or PM treatments

Response 65: We have made correction according to the Reviewer’s comments. We have changed “than that in treatment AM or PM treatment” to “than that in the AM or PM treatments”. In line 358.

Point66: -Figure 7. Need to add more to the methods about how you managed the FAPROTAX and FUNguild assignments. For example, in FUNguild, which confidence levels did you accept? Also what did you do with taxa that were assigned to multiple guilds? In the caption, change “represent” to “represents”.

Response 66:

  • The confidence level is generally classified into three levels: highly probable, probable and possible, and there is no specific value set.
  • The funguild is matched by the annotation information of OTU, and the analysis is performed according to the FUNguild database, which is generally not annotated to more than one guild.
  • We have changed “represent” to “represents”.

[1] Nguyen, N.H.; Song, Z.; Bates, S.T.; Branco, S.; Tedersoo, L.; Menke, J.; Schilling, J.S.; Kennedy, P.G. FUNGuild: An open annotation tool for parsing fungal community datasets by ecological guild. Fungal Ecol. 2016, 20, 241-248.

In line 363.

Point67: -Line 320: was

Response 67: We have made correction according to the Reviewer’s comments. We have changed “were” to “was” In line 380..

Discussion:

Point69: -Line 368: what do you mean general researchers? Do you mean “In general, researchers…”

Response 69: We have made correction according to the Reviewer’s comments. Thank you for pointing out the error. We have changed “General researchers” to “In general, researchers”. In line 429.

Point70: -Line 370: systems

Response 70: We have made correction according to the Reviewer’s comments. We have changed “system” to “systems”. In line 431.

Point71: -Line 370: change A to a and delete “research”

Response 71: We have made correction according to the Reviewer’s comments. We have delete “research”. In line 431-432.

Point72: -Line 403: I would also attribute the increase in diversity to the present of more diverse plant litter and exudates

Response 72: Thank you for the points raised. Since litters and root secretion were not measured in this experiment, the way they were described in the manuscript was changed.

Point73: -Line 409: this is not correct - you said in line 402 that intercropping increased bacterial diversity (which is what Figure 3 shows)

Response 73: We have made correction according to the Reviewer’s comments. We have changed “reduction” to “addition”. In line 472.

Point74: -Line 422: Mouhamadou ref needs to be formatted as the others with a number.

Response 74: We have made correction according to the Reviewer’s comments. Thank you for pointing out the error, we have edited all the references in the manuscript.

Point75: -Line 435: change “become” to “may have been”

Response 75: We have made correction according to the Reviewer’s comments. We have changed “become” to “may have been”. In line 500.

Point76: -Line 436: change “inhibition of” to “decrease in”

Response 76: We have made correction according to the Reviewer’s comments. We have changed “inhibition of” to “decrease in”. In line 501.

Point77: -Line 437: Acidobacteria are

Response 77: We have made correction according to the Reviewer’s comments. We have changed “Acidobacteria is” to “Acidobacteria are”. In line 502.

Point78: -Line 438: change “absent” to “low”

Response 78: We have made correction according to the Reviewer’s comments. We have changed “absent” to “low”. In line 504.

Point79: -Line 456: but why wouldn’t the decrease in SOM also cause a decline in Ascomycota?

Response 79: We are sorry for the misunderstanding caused by our writing error, we have corrected it. We have changed “bacterial” to “Ascomycota”. In line 521.

Point80: -Line 461: change “now” to “arguably” . The makers of PiCRUST and PiCRUST2 would disagree…

Response 80: We have made correction according to the Reviewer’s comments. We are sorry for the misunderstanding caused by our writing error, we have corrected it. We have changed “now” to “arguably”. In line 527.

Point81: -Line 466: Chemoheterotrophy (change to one word, here and elsewhere)

Response 81: We have made correction according to the Reviewer’s comments. We have checked the expressions in the full text to ensure consistency. In line 532-539.

Point82: -Line 469: Chemoheterotrophic bacteria

Response 82: We have made correction according to the Reviewer’s comments. We have changed “Chemo heterotrophy bacteria” to “Chemoheterotrophic bacteria”. In line 536.

Point83: -Line 471: chemoheterotrophic groups

Response 83: We have made correction according to the Reviewer’s comments. We have changed “chemo heterotrophy groups” to “chemoheterotrophic groups”. In line 542.

Point84: -Line 472: change “chemical heterotrophic” to “chemoheterotrophic”

Response 84: We have made correction according to the Reviewer’s comments. We have changed “chemo heterotrophy” to “chemoheterotrophic”. In line 536.

Point85: -Line 473: how do you know there was increased fine root turnover? You did not measure this.

Response 85: We apologize for this point. We have removed the point and added a new point about the function of chemoautotrophic microbes.” In this study, the SOC, TN and SAN contents in the soil were lower in the I mode than in the AM mode, and the closely related aerobic chemoheterotrophy and chemoheterotrophy functional communities were also significantly reduced, further demonstrating the close relationship between the bacterial functional communities and environmental factors”. In line 537-541.

Point86: -Line 480: soil saptrotrophs dominate

Response 86: We have made correction according to the Reviewer’s comments. We have changed “Soil Saprotroph dominates” to “soil saptrotrophs dominate”. In line 551.

Point87: -Line 482: change “intermediate crop” to “intercropping mode, which”

Response 87: We have made correction according to the Reviewer’s comments. We have changed “intermediate crop” to “intercropping mode, which”. In line 554.

Conclusions:  

Point88: -In general, need to tone down the language some, here and in the discussion.

Response 88: We have made correction according to the Reviewer’s comments. Apologies for the errors in our writing, we will make changes in the manuscript.

Point89: -Line 487: despite the fact that

Response 89: We have made correction according to the Reviewer’s comments. We have changed “despite the” to “despite the fact that”. In line 559.

Point90: -Line 487: the intercropping mode

Response 90: We have made correction according to the Reviewer’s comments. We have changed “intercropping mode” to “the intercropping mode”. In line 560.

Point91: -Line 488: We suggest the decrease…( you are suggesting this, you do not know this for sure)

Response 91: We have made correction according to the Reviewer’s comments. We have changed “The decrease” to “We suggest the decrease”. In line 561.

Point92: -Line 490: may have changed the competition (again, it may have done this, but you don’t know for sure)

Response 92: We have made correction according to the Reviewer’s comments. We have changed “changed the competition relationship” to “may have changed the competition”. In line 565.

Point93: -Line 490: change “competition relationship” to “competitive relationships”

Response 93: We have made correction according to the Reviewer’s comments. We have changed “competition relationship” to “competitive relationships”. In line 563-564.

Point94: -Line 492: changed (past tense)

Response 94: We have made correction according to the Reviewer’s comments. We have changed “changes” to “changed”. In line 565.

Point95: -What about plant growth? This is a key piece missing from this study. It would be great to connect all of this soil work back to the important outcome variable, which is the crop production!

Response 95: Thank you for your innovative ideas. In this study, the above-ground and root biomass of the pandan and areca nut will be measured in subsequent experiments, and the mechanism of intercropping regulating crop yield by affecting the interactions between soil, root system and microbes will be investigated.

Some other things to consider:

Point96: -It seems like the authors need to clarify that the enzymes are produced by microbes. Sometimes the authors imply that enzyme activity is affecting microbial communities but it is likely the other way around. Probably the crop management system changes soil properties and microbes and the enzyme activities are in turn affected by this.

Response 96: Thank you for your comments and perspectives. Soil enzymes are produced by the combined action of root secretions and soil microbes.

There is a complex correlation between soil enzymes-soil microbes-soil properties, and soil enzyme activity may have a strong feedback effect on soil for microbes.

Point97: -The authors should comment on the 3 points in Figure 4a that are out to the left (low axis 1 values). Why are these 3 replicates different than the other 3 replicates? It is cool that 3 of the replicates in the I mode fall in between the AM and PM modes in ordination space!

Response 97: Thank you for your comments and perspectives. It may be that there may be some heterogeneity between two more distant points due to the large area in the block.

Round 2

Reviewer 1 Report

Although the revised version has done some corrections and improved its quality to some extent, my major concerns remain unresolved.

1.      I think that the authors may not fully understand what this current experimental design can reveal. According to the title of this manuscript, the authors aimed to know the effect of intercropping Pandanus amaryllifolius with Areca catechu on soil properties and microbial community. Thus, the manuscript is supposed to articulate the advantage and disadvantage of the intercropping mode in soil properties and microbial community comparing with pure Areca catechu plantation, and explain why this happens. However, it seems that the authors prefer discussing the effects of environmental factors on the diversity and functions of soil microbial communities. Question arises! You don’t set up environment gradients, how can the relationships between environment factors and soil relationship or microbial community be clearly assessed?

2.      The authors repeat soil health in the manuscript. But what is soil health? Is intercropping mode conducive to soil health? And how? In addition, the authors did not mention soil health in the introduction section, while soil health is included in the key words. Is it appropriate?

3.      The authors do not set up the experiment to compare intercropping of cash crops with annual crop. Why do you mention this in the introduction?

Author Response

Dear Editors and Reviewers:

Thank you for your letter and for the reviewers’ comments concerning our manuscript entitled “Effects of Intercropping Pandanus amaryllifolius on Soil Properties and Microbial Community Composition in Areca catechu Plantations” (ID: forests-1980905).

Those comments are all valuable and very helpful for revising and improving our paper, as well as providing important guidance for our research. We carefully considered the comments and made changes that we hope will be accepted. The revised sections are highlighted in the paper. The major updates are as follows:

(1) We ensured that the references used in the paper were content-relevant, and we used the Track Changes feature to mark changes to the article so that editors and reviewers could easily see them.

(2) Adopted the reviewer's recommendation to add more details to the methods and analysis.

(3) With many changes, we reorganized some of the structure of the article as suggested by the reviewers, particularly the Results and Discussion sections. We also proofread the manuscript and carefully checked and corrected any grammatical errors. Please find below specific clarifications/changes to questions/comments raised by reviewers in response to this cover letter.

We tried our hardest to improve the manuscript and made some changes. These modifications will have no effect on the paper's content or structure. We sincerely thank the Editors/Reviewers for their hard work and hope that the correction is accepted.

I am looking forward to hearing from you soon.

Kind regards,

Yiming Zhong

2022-10-25

Reviewer 2 Report

The authors have made many of the necessary corrections, but there are still several items to address.

Line 28: FUNGuild not Funguild

Line 147: For the correction here, sorry I was not clear. I mean replace “and the block was repeated 6 times” to “Each block was replicated 6 times”. Keep the text that says “Each block had three plots”

Lines 195-200: Need to cite the papers that presented FLASH, QIIME, MOTHUR, SILVA, and UNITE. The people who developed those resources deserve credit!

Line 198: Just because OTU is still used is not a good justification to use it instead of ASVs. I suggest the authors use ASVs in the future. However, at this stage it is not necessary to rerun everything with ASV instead of OTUs, but for future projects the authors should consider this.

Line 201: normalization method. instead of “subsample method”, it would be more clear if you just said that you rarefied the data. You should also state how many sequences per sample the data were rarefied to. I do not agree with or perhaps just don’t understand the authors’ response on this point. There is indeed sequencing depth for microbial marker gene sequencing projects (e.g. Caporaso et al. 2010).

Line 211: Again, cite the papers associated with these programs so those authors get credit for developing great tools for us to use. FAPROTAX would be Louca et al. 2016 Science. FUNGuild would be Nguyen et al. 2016 Fungal Ecology.

Line 212: Need to actually add details to the text about FUNGuild methods, not just in response to me. Clarify that you used highly probable, probable, and possible (this is important to point out, because other authors, for example, do not retain the “possible” category. I am fine with any method as long as it is clearly stated so readers can evaluate what was done). In your response, you stated that “generally” there was only one guild. That implies that in some cases there were more than one guild applied and thus the authors should still state in the text how this was managed. I have worked with FUNGuild before and have seen this a lot.

Line 213: If you used 338f and 806r primers, it is possible that they may have amplified some archaeal DNA. Did you check your taxonomic table for archaea? If not, please do so and confirm in the text that there were no archaeal taxa in the 16S dataset. 

Line 215: You still did not mention PERMANOVA in the methods. This must be added to the methods text in the statistical analysis section. 

Line 403: I don’t agree with your response and it looks like you have not added anything about litter or exudates here. I really think this deserves some mention, and it doesn’t matter if you did not measure it. This is the discussion, and you should be raising some possible explanations for the data even if they weren’t measured.

Point 95: In the conclusions, perhaps you should mention in the conclusions that ongoing and future work will test how this will affect plant growth. I am glad the authors are pursuing this and look forward to seeing their future results.

Figures:

Nice updates on the Figures. A few more changes are needed:

Figure 4: I like the new combined figure, but I suggest making the vectors and text slightly transparent so they do not dominate the figure so much. And perhaps make the sample data points a bit bigger.

Figure 7: Need to show a bar for each individual sample (just as the new Figure 5, which looks great!). Sorry I did not mention that in the first round.

Figure A1 caption: clarify that F and P values are from PERMANOVA. Also I believe that is a “pseudo-F” value technically.

Author Response

(The authors gave the same response as above.)
